**Measurement report: Diurnal and temporal variations of sugar compounds in suburban aerosols from the northern vicinity of Beijing, China: An influence of biogenic and anthropogenic sources**

**Santosh Kumar Verma,**[1,2] **Kimitaka Kawamura,**[1,3*] **Fei Yang,**[1,4] **Pingqing Fu,**[1,5] **Yugo Kanaya,**[6] **and Zifa Wang**[7]

[1]Institute of Low Temperature Science, Hokkaido University, Sapporo 060-0819, Japan

[2]State Forensic Science Laboratory, Home Department, Government of Chhattisgarh, Raipur 491001, India

[3]Now at Chubu Institute for Advanced Studies, Chubu University, Kasugai 487-8501, Japan

[4]Graduate School of Environmental Science, Hokkaido University, Sapporo 060-0810, Japan

[5]Institute of Surface-Earth System Science, Tianjin University, Tianjin 300072, China

[6]Research Institute for Global Change, Japan Agency for Marine-Earth Science and Technology, Yokohama, Japan

[7]LAPC, Institute of Atmospheric Physics, Chinese Academy of Sciences, Beijing, China

*Corresponding author:* Kimitaka Kawamura

E-mail: kkawamura@isc.chubu.ac.jp

Key points:

1. Autumn time observations of sugar compounds (SCs) in the northern vicinity of Beijing, China.
2. Influence of natural biogenic emissions on SCs from forest area.
3. Influence of anthropogenic and bioaerosol on SCs from the Beijing area.
4. Biomass burning is a significant contributor to SCs.
5. Biogenic and fungal-microbial emissions are significant sources for mannitol and arabitol.

**Abstract**

Sugar compounds (SCs) are major water-soluble constituents in atmospheric aerosols. In this study, we investigated their molecular compositions and abundances in the northern receptor site (Mangshan) of Beijing, China, to better understand the contributions from biogenic and anthropogenic sources using a gas chromatography–mass spectrometry technique. The sampling site receives anthropogenic air mass transported from Beijing by southerly winds, while northerly winds transport relatively clean air mass from the forest areas. Day- and nighttime variations were analyzed for anhydrosugars, primary sugars, and sugar alcohols in autumn 2007. We found that biomass burning (BB) tracers were more abundant in nighttime than daytime, while other SCs showed different diurnal variations. Levoglucosan was found as a dominant sugar among the SCs observed, indicating an intense influence of local BB for cooking and space heating at the surroundings of the Mangshan site. The high levels of arabitol and mannitol in daytime suggest a significant contribution of locally emitted fungal spores and long-range transported bioaerosols from the Beijing area. The plant emissions from Mangshan forest park significantly control the diurnal variations of glucose, fructose, and mannitol. The meteorological parameters (relative humidity, temperature, and rainfall) significantly affect the concentrations and diurnal variations of SCs. Sucrose (pollen tracer) showed a clear diurnal variation, peaking in the daytime due to higher ambient temperature and wind speed, which influences the pollen release from the forest plants. We found the contribution of trehalose from soil dust in daytime, while microbial and fungal spores were responsible for nighttime. Anhydrosugar and primary sugars are prime carbon sources of the Mangshan aerosols. The high ratios of levoglucosan in organic carbon and water soluble organic carbon in nighttime suggest a significant contribution of BB to organic aerosols at night. Levoglucosan/mannosan ratios demonstrate that low temperature burning of hardwood is dominant in Mangshan. The positive matrix factorization analysis concluded that forest vegetation, fungal species, and local BB are the significant sources of SCs.

**Keywords:** Anthropogenic bioaerosols, biomass burning, pollen tracer, fungal tracers, soil dust, and microbial tracers

## 1. Introduction

Increased economic growth and massive consumption of fossil fuels from industries emit anthropogenic gases, aerosols, and biomass burning (BB) products cause severe air pollution in East Asian countries (Lelieveld et al., 2015; Lin et al., 2014; Kawamura et al., 2013; Li et al., 2010; Sun et al., 2016). Globally, significant anthropogenic and carbonaceous aerosols are contributed by China (Cooke et al., 1999, Wang et al., 2007). Beijing is situated in the northern part of China, with a 20 million people and 5 million motor vehicles. Beijing is one of the largest polluted cities in East Asia; its air quality deteriorates seriously due to massive emissions of anthropogenic aerosols from vehicles and industries (Cao et al., 2014; Qiao et al., 2018; Tao et al., 2017; Wei et al., 2018; Yu et al., 2013). Organic aerosols (OAs) are composed of a complex mixture of diverse molecules (Xu et al., 2011). They play essential roles in global climate changes via the modification of radiative forcing and cause a serious negative impact on human health (Fuzzi et al., 2007). OAs contain various water-soluble organic compounds, which can act as cloud condensation nuclei (CCN) (Kanakidou et al., 2005).

BB is essentially a primary source of OAs, controlling the air quality levels and affecting the earth's radiative forcing by scattering or absorbing incident solar radiation (Deshmukh et al., 2019a; Kanakidou et al., 2005; Kanaya et al. 2013; Streets et al., 2003; Sullivan et al., 2008). There are several kinds of BB, including industrial biofuel burning, open field burning (fires of the forest, peatlands, and agricultural wastes), and domestic BB burning for house heating and cooking, which emits BB products into the atmosphere (Akagi et al., 2011). The BB aerosols are subjected to long-range atmospheric transport once they are emitted into the atmosphere (Verma et al., 2015). Levoglucosan (1,6-anhydro-β -D-glucopyranose) is a pyrolysis product of cellulose and hemicellulose, which is generally found as major organic constituents in the BB-influenced aerosols (Simoneit et al., 1999; 2002). Levoglucosan have been reported as a specific tracer for BB aerosols (Engling et al., 2009).

Sugar compounds (SCs) are ubiquitous in the atmosphere from different geographical locations, including urban, forest, marine, and polar regions (Burshtein et al., 2011; Fu et al., 2010; Wan et al., 2017). SCs are emitted from algae, microbes, pollen, suspended soil particle, and associated biota into the atmosphere by various processes, and thus they are termed as primary biological aerosol particles (PBAPs) (Carvalho et al., 2003; Despres et al., 2012; Elbert et al., 2007). The detailed study of bio-aerosols has been emphasized in the past decades due to the global impact of microbes and fungi because they can travel long distances from the source regions by winds (Burshtein et al., 2011; Brown and Hovmoller, 2002; Yamaguchi et al., 2012).

Fungi are essential microbes in the ecosystem, which discharge spores of 8-186 Tg yr$^{-1}$ into the
atmospheric environment (Elbert et al., 2007; Heald and Spracklen, 2009). Sugar alcohols like
arabitol and mannitol are enriched in fungal spores; thus, they are considered as specific tracers
(Bauer et al., 2008).
Devis et al. (1988, 1990) reported that mannitol was also found in about 70 different
higher plant families. Loescher et al. (1992) reported that mannitol is an important photosynthetic
product converted by biosynthesis in plants. Keller and Matile (1989) also found the arabitol and
mannitol during the increased photosynthesis in growing vegetation. Pollens are the largest
particles that could contribute up to 65% of the PBAPs, which are the significant sources for
sucrose and fructose in the forest aerosols (Manninen et al., 2014; Pacini, 2000). Higher plants
synthesize primary sugars (glucose, fructose, and sucrose) during photosynthesis, which are
circulated by phloem to accumulate in root cells and to develop plant sections (Jaenicke, 2005; Jia
et al., 2010; Pacini, 2000). Cowie et al. (1984) also reported various sugars in terrestrial plant
fruits, flowers, and plant tissues. Bieleski (1995) reported that glucose, fructose, and sucrose are
well-known components of microbes and invertebrates. The plant debris, as well as lichens,
invertebrates, and soil dust, are also recognized as possible sources of primary sugars in the
atmosphere (Medeiros et al., 2006; Rogge et al., 2007; Simoneit et al., 2004).
Previous studies analyzed aerosol samples for SCs and discussed several factors to control
their local and global atmospheric levels. Recently, Xu et al. (2020) examined the seasonal
molecular distributions of primary biological aerosols and BB aerosol samples collected from
urban Beijing. They reported a high level of arabitol, mannitol, sucrose, glucose, and fructose in
the vegetation-growing season. Kang et al. (2018) also reported higher concentrations of sugars in
the urban aerosols from Beijing. They suggested a large contribution of coal combustion and
agriculture residue burning under stable meteorological conditions in winter and spring. Verma et
al. (2015, 2018) reported that the atmospheric circulations and long-range transport of organic-
/bio-aerosols from East Asia significantly control the levels and compositions of SCs over the
western North Pacific. The above studies discussed the several factors that affect the
concentrations of SCs in the aerosol samples collected from urban and remote areas.
In this study, we conducted analyses of SCs in the aerosol samples collected from the
northern vicinity of Beijing City in 2007. Here, we present comprehensive data sets of
anhydrosugars, primary sugars, and sugar alcohols in the suburban aerosol samples and their
diurnal variations to explain the source variance following the wind patterns in the day- and
nighttime. The positive matrix factorization (PMF) has been applied to clarify the different
sources of measured SCs in the aerosol. We present the influence of local meteorology of
sampling site and atmospheric transport from Beijing by southerly winds and Mangshan National
Forest Park by northerly winds on the molecular distributions of SCs. Using the mass
concentration ratio of levoglucosan to mannosan, we explain the relative contribution of hard and
softwood burning to the air quality of Mangshan. This study also discussed carbon contributions
of SCs and BB measured in the Mangshan aerosol samples from different sources.
**2. Materials and Methods**
**2.1. Site description and aerosol sample collection**
The sampling site (Mangshan: 40.28 N, 116.26 E) is located 40 km north of Beijing. A
detailed description of the sampling site is given in He et al. (2014, 2015). Briefly, Mangshan is
surrounded by urban areas in the south and forest areas with the national park in the north (Fig. 1).
The ambient temperature was higher in daytime (23.9°C) than nighttime (12.1°C), with an average
of 17.8°C during the campaign. The relative humidity (RH) varied significantly from 22.1% to
90.5%, with an average of 51.7% during the study period. The rainfall was observed at midnight
on 15[th] September, the morning of 17[th] to evening 18[th] September, the night of 26[th] September,
and light rain lasted from 4[th] October to the end of the campaign (Fig. 2a). Interestingly, the
sampling site is characterized by a specific wind pattern, i.e., southwest wind (69.9%) prevailed,
followed by northeast wind (23.4%) and southeast wind (6.2%) during the daytime (Fig. 2a). The
northeast wind (99.5%) was dominated at night, which is consistent with the air mass back
trajectories (He et al., 2014) (Fig. 2b). The daytime wind from the southwest direction passed over
Beijing, delivering anthropogenic air mass to the Mangshan site.
Detailed descriptions of the total suspended particulate (TSP) samples collected at
Mangshan are given in He et al. (2014, 2015). Briefly, The aerosol samples were collected near
the entrance of Mangshan National Forest Park. The elevation of the sampling location is 187 m
above sea level. A high-volume air sampler (Kimoto-AS810A) at a flow rate of 1.13-1.17 $m^3$ min[-]
[1] was used to collect the TSP without cut-off device. In the sampling, no denuder was applied to
remove semi-volatile gases because the filter samples were used to analyze nonvolatile sugar
compounds. However, the levoglucosan partition between the gas and particle phases, but their
concentration was low. The sampling time was rather short due to the day and night sampling.
Therefore, the uncertainty due to the gas phases in the particulate species concentration might be
insignificant. The samples were collected on pre-combusted (450°C for 6 h) quartz fiber filters
(Pallflex 2500QAT-UP, 20 cm × 25cm) from 15[th] September to 5[th] October 2007. After sample

collection, the individual filters were placed in pre-combusted glass jars with Teflon-lined screw
caps and stored in a dark, cold room at −20°C to prevent microbial activity and loss of semi-
volatile organic compounds from the samples. In this study, a total of 58 filter samples were
analyzed. We collected 3h daytime (from 9 to 12, 12 to 15, 15 to 18 h) (n=26), 9h daytime (from 9
to 18 h) (n=12), and 15h nighttime (from 18 to 9 h) (n=20) samples together with four field
blanks. Table S1 shows the details of aerosol sample collection in the Mangshan site.

## 2.2. Extraction and derivatization of samples

A total of 58 aerosol samples were analyzed for anhydrosugars, primary sugars, and sugar
alcohols (Table 1). The sample filters (approximately 21 cm$^2$) were extracted with a
dichloromethane and methanol mixture (2:1) under ultrasonication. Pasteur pipettes packed with
pre-combusted quartz wool were used to filter the extracts to remove filter debris. After filtration,
the extracts were concentrated in a rotary evaporator under vacuum and dried by nitrogen
blowdown. The extracts were reacted with 60 µl of N,O-bis-(trimethylsilyl)trifluoroacetamide
(BSTFA) with 1% trimethylsilyl (TMS) chloride in the presence of 10 µL of pyridine at 70°C for
three hours to derivatize hydroxyl (OH) and carboxyl (COOH) groups into corresponding
trimethylsilyl (TMS) ethers and esters, respectively,  After the reaction, n-hexane was used for
dilution, and C$_{13}$ n-alkane was added as an internal standard before GC-MS analysis.

## 2.3. Gas chromatography-mass spectrometry determination of sugar compounds (SCs)

Details of GC-MS operation and identification of SCs are described in Verma et al. (2015,
2018). Briefly, GC-MS analyses were performed on Agilent model 6890 gas chromatograph (GC)
combined with an Agilent model 5973 mass selective detector (MSD) to determine SCs. The mass
spectrometer was operated in the electron ionization (EI) mode at 70 eV with a scan range of $m/z$
40–650. The GC separation was achieved on a DB-5MS fused silica capillary column (30 m ×
0.25 mm in diameter, 0.25 μm film thickness) and a split/splitless injector. The GC oven
temperature was programmed to maintain at 50$^{\circ}$C for 2 min and then to increase from 50 to 120$^{\circ}$C
at a rate of 15$^{\circ}$C min$^{-1}$, then from 120 to 305°C at a rate of 5°C min$^{-1}$. The final isotherm holds at
305$^{\circ}$C for 15 min. Helium was used as the carrier gas at a flow rate of 1.0 mL min$^{-1}$. The sample
was injected on a splitless mode at 280$^{\circ}$C injector temperature. GC-MS data were acquired and
processed with Agilent GC/MSD ChemStation software.

The individual compounds (TMS derivatives) were identified by comparing the relative
response factors determined by the injection of authentic standards and those reported in the

literature and library texts (Claeys et al., 2004). Fragment ions of sugar compounds at 217 and 204 were used for quantifications. Total ten sugar compounds, including three anhydrosugars (levoglucosan, galactosan, mannosan), four primary sugars (glucose, fructose, sucrose, trehalose and xylose) and three sugar alcohols (arabitol, mannitol, and inositol), were detected in the Mangshan aerosols. Field blanks were treated as a real sample and analyzed by the procedure used for the real samples. Recoveries for SCs were better than 85% as obtained by the standards spiked to precombusted quartz filter followed by extraction and derivatization. Based on the duplicate analysis, the analytical errors in the concentrations of the detected compounds were obtained to be within 10%. The detection limits of SCs corresponds to ambient concentrations of 150-620 pg $\mu$L$^{-1}$, which corresponds to ambient concentrations of 15-70 pg m$^{-3}$ under a typical sampling volume of 900 m$^3$.

## 2.4. Chemical analyses of organic carbon (OC), water-soluble organic carbon (WSOC), and inorganic ions

The data set and methods for the determination of organic carbon (OC), water-soluble organic carbon (WSOC) and inorganic ion (Ca$^{2+}$) were reported in He et al. (2015). Briefly, the concentrations of OC were measured using a semi-continuous OC/EC analyzer (Sunset Laboratory Inc., Portland, OR, USA). A punch of the filter ($\Phi$14 mm) was placed in a quartz boat inside the thermal desorption chamber of the analyzer, and then stepwise heating (IMPROVE) was applied. The oven temperature was programmed as follows: under He, every 2 minutes, the oven temperature was increased starting from 250°C for 2 min, at 450°C for 2 min, and at 550°C for 2 min. After that, 550°C was maintained for two minutes under He mixed with 10% O$_2$, then at 700°C for 2 min and at 870°C for 3.5 min. NDIR detector was used to determine CO$_2$ generated in the above process (Wang et al., 2005b). The carbon content of the sample that evolves to CO$_2$ between 250 and 700°C was defined as OC.

Aliquots of the filter samples (3.14 cm$^2$) were extracted with Milli Q water for the water-soluble inorganic ion and WSOC measurements. After extraction, one part was used for the analyses of inorganic ions (SO$_4^{2-}$, NO$_3^-$, Cl$^-$, NH$_4^+$, Na$^+$, Ca$^{2+}$, K$^+$ and Mg$^{2+}$) using an ion chromatography (IC) system (761 Compact IC, Metrohm, Switzerland). Cations on a Shodex YK-421 column with 4mM H$_3$PO$_4$ as eluent and anions were separated on a Shodex SI-90 4E column with 1.8mM Na$_2$CO$_3$ and 1.7mM NaHCO$_3$ as eluent. The injection loop volume was 200 μl. Both cations and anions were quantified against a standard calibration curve. Another part of the filtered water extract was acidified with 1.2 M HCl and purged with pure air to remove dissolved inorganic carbon and volatile organics. Then WSOC was measured with a carbon analyzer

(Shimadzu, TOC-5000). Procedural blanks were carried out in parallel with real samples to account for any contamination (He et al., 2015).

**2.5. Positive Matrix Factorization (PMF) Analysis**

Positive matrix factorization (PMF) is a powerful statistical tool for resolving the potential sources contributing to atmospheric particles (Paatero and Tapper, 1994). The measured ambient concentrations and method detection limits (MDLs) of SCs were used to calculate the uncertainties. The measured concentrations of SCs below or equal to the MDLs were replaced by half of the MDL, and associated uncertainties were set at 5/6 of the MDL [(5/6) × MDL] values of each sample. The geometric mean concentrations were used for missing concentrations, and the uncertainty of the concentrations greater than the MDL was calculated based on the following equation:

$$\text{Uncertainty} = \sqrt{(\text{error fraction} \times \text{concentration})^2 + (0.5 \times \text{MDL})^2}$$

The error fraction is a user-provided estimation of the analytical uncertainty of the measured concentration or flux. For example, Han et al. (2017) used an error fraction of 0.2-0.3 for organics and 0.2 for all the species. In this work, the error fraction was set to be 0.3 for all species. Paatero et al. (2002) and Zhou et al. (2004) reported detailed discussions of the determination and application of PMF analysis.

**3. Results and Discussion**

**3.1. Ambient concentrations and diurnal variations of SCs**

We detected a total of ten SCs, including three anhydrosugars, four primary sugars, and three sugar alcohols in the Mangshan aerosol samples. Figure 3a-c showed the temporal variations and Table 1 showed minimum, maximum, and average concentrations of anhydrosugars, primary sugars, and sugar alcohols with a standard deviation. The overall concentrations of SCs varied from 30.8–875 ng m$^{-3}$ (avg. 325 ng m$^{-3}$), which was higher in the daytime (315 ng m$^{-3}$) and lower at nighttime (276 ng m$^{-3}$), however, we did not observe statistically significant differences (student t-test, 95% confidence interval, p > 0.05) in their atmospheric abundances. Interestingly, higher average concentrations of SCs were reported for the aerosol samples collected from at Mt. Tai (daytime 640 ng m$^{-3}$ and nighttime 799 ng m$^{-3}$) in the North China Plain (Fu et al., 2008) than the Mangshan aerosol. The diurnal concentrations of SCs may be significantly influenced by vegetation and BB activities in the Mangshan site. SCs are significantly contributed by plant

fractions and fungus from the forest area (Zhu et al., 2016). The meteorological parameters also
affect the concentrations of SCs in the forest site (Miyazaki et al., 2012).
In addition, anthropogenic aerosols emitted from urban areas are probably transported to the
northern receptor site in daytime by a southerly wind (He et al., 2014; 2015). Therefore, the high
levels of SCs in daytime may be related to the transport of organic and bio-aerosols from urban
regions. The nighttime, the wind direction is shifted to northerly, delivering comparatively clean
air masses from the Mangshan National Forest area to the sampling site. Air mass from the forest
may significantly contribute to nighttime SCs in the Mangshan site. The influence of local sources
and long-range transported aerosols on the SCs will be discussed in sections 3.1.1 to 3.1.3.
**3.1.1. Ambient concentrations and diurnal variations of anhydrosugars**
The average concentrations of anhydrosugars were found 116 ng m$^{-3}$, contributing 31.9%
of overall SCs in the Mangshan aerosols (Table 1). Figure 4a-c shows the temporal variations of
anhydrosugars. They are more abundant in nighttime (avg. 152 ng m$^{-3}$) than daytime (avg. 97.1 ng
m$^{-3}$). Levoglucosan (100 ng m$^{-3}$) is the most abundant anhydrosugar followed by galactosan (10.1
ng m$^{-3}$) and mannosan (6.05 ng m$^{-3}$) detected in Mangshan aerosols. Kang et al. (2018) reported
high levels of levoglucosan (avg. 110 ng m$^{-3}$) in autumn aerosols from Beijing, China. It is well
known that biofuel burning is the common energy source for cooking and house heating in China
in winter and autumn (Verma et al., 2015), thus the domestic BB activities in the surroundings of
the Mangshan site significantly contribute to the levoglucosan. BB tracers showed significant
positive correlations with each other (levoglucosan and galactosan, r = 0.98; levoglucosan and
mannosan, r = 0.97; galactosan and mannosan, r = 0.98), suggesting their similar sources in the
Mangshan aerosols (Table 2).
The levoglucosan concentrations showed significant diurnal variations, which was higher
in nighttime (avg. 132 ng m$^{-3}$) than daytime (avg. 83.2 ng m$^{-3}$) (Table 1). A similar diurnal pattern
was also found for the concentrations of galactosan and mannosan. The increased concentrations
of BB tracers were observed during the periods of lower ambient temperature (Figs. 2a, 4a-c). The
higher ambient temperature was recorded in daytime between 09h to 15h during the campaign,
associated with declined BB activities. In this sequence, the nighttime samples were collected
from 18:00h to 09:00h, including peak hours of BB for domestic purpose. Therefore, it is
reasonable to detect higher abundances of BB tracers in the nighttime than daytime. Hence, it is
evident that BB activities were increased at night because of cooking and house heating at cool
night in autumn. In addition, recent studies reported the widespread BB aerosols in the North
China Plain, including megacities such as Beijing, Nanjing, Hebei, and Tianjin (Lelieveld et al.,
2015; Kawamura et al., 2013; Li et al., 2010; Sun et al., 2016). Therefore, the atmospheric
transport of BB aerosols from the urban area to the Mangshan site by southerly winds cannot be
excluded. The diurnal variations of levoglucosan may be significantly influenced by the local BB
activities and transported BB aerosols from urban areas, where BB products are generated by
brown coal combustion (Yan et al., 2018).

### 286    3.1.2. Ambient concentrations and diurnal variations of primary sugars

The fragment of vascular plants contains primary sugars, including glucose, fructose,
sucrose, and trehalose (Medeiros et al., 2006). Primary sugars were found as the most abundant
sugars (avg. 133 ng m$^{-3}$), contributing to 41.8% of the total SCs in Mangshan aerosols (Table 1).
They showed apparent diurnal variations with daytime high (avg. 166 ng m$^{-3}$) and nighttime low
values (avg. 69.4 ng m$^{-3}$) (Figs. 3a-c, 5a-d). Graham et al. (2003) also reported similar diurnal
variations of primary sugars for the Amazon forest aerosols. Sucrose was found as dominant
primary sugars (avg. 58.5 ng m$^{-3}$), accounting for 44% of measured primary sugars in Mangshan
aerosols (Table 1). Pollen was reported as a primary source for sucrose in aerosols collected from
a Texas rural site (Jia et al., 2010). Fu et al. (2012) found high sucrose concentrations up to 1390
ng m$^{-3}$ in the aerosols from Jeju Island, South Korea. Therefore, the plant materials, including
pollen spores from the local vegetation of Mangshan National Forest Park, are likely the primary
source of sucrose in the aerosols. Miyazaki et al. (2012) also reported higher sucrose
concentrations in the aerosol samples collected from the Hokkaido deciduous forest.
We found a significant diurnal variation of sucrose with higher daytime (82.9 ng m$^{-3}$) than
nighttime (12.3 ng m$^{-3}$). Meteorological parameters such as temperature, rainfall, wind speed, and
solar radiation significantly influence pollen activities and, subsequently, sucrose concentrations
(Verma et al., 2018). Interestingly, an elevated peak of sucrose was observed from 12h to 15h with
higher ambient temperature. In contrast, lower sucrose concentrations were observed from 15h to
9h with lower ambient temperature (Fig. 5a). Daytime increased concentrations of sucrose might
be related to the higher daytime ambient temperature, low RH, and high solar radiation (Miyazaki
et al., 2012). Taylor et al. (2002) reported the influence of the meteorological conditions, i.e.,
strong daytime winds and convective activity, which can result in catapulting of pollen, opening of
pollen-laden flower anthers, and causing enhance entrainment and dispersal of the particles into
the air. Pacini (2000) reported that higher levels of sucrose in daytime coincide with higher counts
of pollen, fern spore, and insect. The positive linear correlations of sucrose with ambient
temperature (r = 0.52) and solar radiation (r = 0.55) further supported the influence of

meteorological parameters in the sucrose concentration (Table 2).

Five rain events were recorded during the campaign, i.e., 15[th], 17[th], 18[th], and 26[th] September, and 1[st] and 5[th] October (Fig. 2a). Pollens are significantly settled down by wet scavenging during rain events because their sizes are large. A low concentration of sucrose was found from the beginning of sampling to the morning of 20[th] September and from the afternoon of 26[th] September to the end of the sampling campaign (Fig. 5a). In addition, the increased concentrations of sucrose were found in the aerosol samples collected from 20[th] to 22[nd] September, and moderate concentrations were observed after 23[rd] to the evening of 25[th] September during non-precipitation events. Consequently, the pollens were significantly scavenged during wet precipitation and washout effect from the atmosphere, resulting in lower sucrose concentrations at the earlier periods, than later periods. In addition, Rogge et al. (2007) reported that surface soil dust and unpaved road dust also contribute sucrose in the atmospheric aerosols. However, insignificant correlations between sucrose and $Ca^{2+}$ (daytime, r = 0.32; night time, r = 0.37) do not supports soil dust contributions to sucrose in the Mangshan aerosols (Table 2).

Glucose was the second dominant primary sugar in the Mangshan aerosols. The average concentrations of glucose and fructose were observed to be 40.0 ng m$^{-3}$ and 20.1 ng m$^{-3}$, respectively (Table 1, Fig. 5b). The sampling site is characterized by the dense vegetation in the Mangshan National Forest Park. Therefore, the nectars and fruits of vegetation (Baker et al., 1998), plant debris (Medeiros et al., 2006) and pollens (Fu et al., 2012) in the forest significantly contribute to glucose and fructose. The glucose levels are equivalent to that (50.1 ng m$^{-3}$) reported from the Howland Experimental Forest site in the USA (Medeiros et al., 2006). Glucose and fructose showed significant diurnal variations, whose concentrations were higher in daytime (44.2 ng m$^{-3}$ and 23.9 ng m$^{-3}$, respectively) than nighttime (32.0 ng m$^{-3}$ and 12.8 ng m$^{-3}$, respectively) in Mangshan aerosols (Table 1, Figs. 3b, c; 5b, c). This diurnal variation could be involved with emissions of pollens, fern spores, and other giant particles by strong winds (Graham et al., 2003; Pacini, 2000). Similar trends of glucose and fructose were reported in the Amazon forest, being coincided with plant fragments and insects (Graham et al., 2003). The autumn decay of vascular plant leaves in the Mangshan forest may have contributed to the levels of glucose and fructose.

Although, the daytime southerly winds deliver anthropogenic air masses from megacities to the sampling site. The daytime winds from the northeast direction (23.4%) also carry air masses from the forest region, transporting primary sugars to the Mangshan site. However, 99.5% of the nighttime hours, the wind is shifted to northeasterly, i.e., in forest region (He et al., 2015), but the emissions of primary sugars at night in the form of plant fragments are lower than in daytime.

Because the daytime ambient temperature and solar radiations significantly induce the emissions
of sugar compounds in the forest site (Miyazaki et al., 2012). Therefore, low glucose and fructose
levels were found at nighttime than daytime aerosols at the Mangshan site (Table 1, Fig. 3).
Previous studies have reported lichens (Dahlman et al., 2003) and soil dust (Nolte et al., 2001;
Rogge et al., 2007) as significant sources of both primary sugars. The concentration of glucose
was insignificantly correlated with soil tracer ($Ca^{2+}$) in day (r = 0.02) and nighttime (r = 0.27),
denying their soil dust contributions in Mangshan aerosol samples.
Trehalose in the environment is significantly controlled by the activities of bacteria, fungi,
yeast, algae, invertebrates, and plant species, as well as suspended soil particles (Medeiros et al.,
2006, Rogge et al., 2007). The average concentration of trehalose was found 14.3 ng m$^{-3}$ (Table 1,
Fig. 5d). Yttri et al. (2007) reported higher trehalose concentrations in the aerosol samples
collected from urban (29 ng m$^{-3}$) and suburban (27 ng m$^{-3}$) than rural (3.8 ng m$^{-3}$) areas in
Norway. The above results emphasize that fungi and microbes associated with anthropogenic and
bioaerosols, emitted in the urban and suburban areas, might be responsible for the trehalose
concentration in aerosol samples (Verma et al., 2018). Trehalose showed insignificant diurnal
variation, whose day and night concentrations were observed 15.3 ng m$^{-3}$ and 12.3 ng m$^{-3}$,
respectively, indicating its different emission sources in day and night for Mangshan aerosols (Fig.
3b, c; 5d).
The southerly winds might transport fungi and microbes associated with bioaerosols, eject
spores under favorable meteorological conditions (high RH and low temperature) (Jones and
Mitchell et al., 1996). Several microbes and fungi discharge spores at nighttime due to high RH
conditions (Ibrahim et al., 2011; Kim and Xiao, 2005; Malik and Singh, 2004; Sharma and Razak,
2003). Interestingly, trehalose is more significantly correlated with arabitol and mannitol (r = 0.76
and 0.85, respectively) in nighttime than daytime (r = 0.49 and 0.51, respectively) (Table 2),
suggesting that fungal and microbial spores contributed to high levels of trehalose in nighttime.
Hackl et al. (2000) found trehalose as dominant sugar in spring aerosols and proposed it as a tracer
for soil dust particles. Trehalose concentration was more significantly correlated with $Ca^{2+}$ (r =
0.82) in daytime than nighttime (r = 0.61), indicating soil dust contribution (Table 2). Therefore,
we hypothesized that winds transported soil particles from the urban area in daytime due to the
active building constructions (He et al., 2015), contributing to the high levels of trehalose in
daytime.
**3.1.3. Ambient concentrations and diurnal variations of sugar alcohols**
The average concentrations of sugar alcohols were found 75.8 ng m$^{-3}$, contributing 26.4%
of total SCs measured in Mangshan aerosols (Table 1). Sugar alcohols showed clear diurnal
variations in daytime high (avg. 87.4 ng m$^{-3}$) and nighttime low (avg. 53.7 ng m$^{-3}$) (Table 1).
Mannitol was found as the dominant sugar alcohol (avg. 44.1 ng m$^{-3}$), followed by arabitol (avg.
29.1 ng m$^{-3}$) and inositol (avg. 2.62 ng m$^{-3}$) (Table 1; Fig. 6a-c). Mannitol and arabitol are
common polyols detected in green algae, lichens, and fungal spores (Bieleski, 1995, Dahlman et
al., 2003; Filippo et al., 2013; Lewis and Smith, 1967; Yttri et al., 2007). Previous studies have
reported that arabitol and mannitol are key components of fungal spores, and thus they are
considered as fungal tracers (Bieleski,1995; Lewis and Smith, 1967). Several fungal and microbial
species released spores during biological activities into the atmosphere (Dahlman et al., 2003;
Bauer et al., 2008; Filippo et al., 2013). Therefore, the autumn time fungal and microbial species
significantly contribute to arabitol and mannitol in the Mangshan aerosol samples.
However, mannitol and arabitol showed a strong positive linear correlation (r = 0.81),
which suggested common origins as reported in earlier studies (Fu et al., 2012) (Table 2). In
contrast, the higher concentration of mannitol than arabitol suggested it had sources in addition to
fungal spores in the Mangshan forest site. In this sequence, several previous studies have
confirmed the significance of mannitol in plant photosynthesis (Loescher et al., 1992; Keller and
Matile, 1989; Rumpho et al., 1983). Pashynska et al. (2002) reported that detritus of mature leaves
can emit mannitol into the atmosphere by wind action. Heald and Spracklen (2009) also found a
correlation between the atmospheric water vapor with mannitol concentrations and leaf area index.
They suggested that the activities of the terrestrial biosphere widely affect mannitol concentrations
in the air. Our PMF results also indicated the substantial contribution of mannitol for vegetation
factor (24.8%), which supports that mannitol is attributed by vegetation from the forest area
(section 3.2).
In addition, the meteorological parameters, including high RH and temperature affect the
fungal and bacterial activities (Kim and Xiao, 2005; Sharma and Razak, 2003). The maximum
growth of fungi and bacteria was observed at 92–100% RH (Ibrahim et al., 2011). Interestingly,
the concentrations of arabitol and mannitol gradually increased after the end of precipitation,
following the increases in ambient temperature and RH (Figs. 2a, 6a, b). Miyazaki et al. (2012)
also discussed the increased contributions of arabitol and mannitol with daytime ambient
temperature and solar radiation in the aerosol samples collected from the forest area. Similar
temporal trends and positive linear correlations were observed between arabitol (r = 0.69) and
mannitol (r = 0.57) with RH, which supports the above phenomenon for Mangshan aerosols
(Table 2). Therefore, we propose that a favorable meteorological condition in autumn increases
the emissions of fungal spores and fragments of forest vegetation, which may be responsible for
arabitol and mannitol contributions in the Mangshan aerosols.

The diurnal variation of mannitol and arabitol were characterized by higher in the daytime

(51.7 ng m$^{-3}$ and 32.5 ng m$^{-3}$, respectively) than nighttime (29.6 ng m$^{-3}$ and 22.5 ng m$^{-3}$,
respectively) (Fig. 3b, c). Yamaguchi et al. (2012) reported that fungal spores and bacterial cells
associated with bioaerosols could be transported long distances. The Mangshan site receives
significant anthropogenic and bioaerosols from Beijing City by southerly winds. Therefore, the
daytime plant activities, influenced by solar radiation and ambient temperature and the long-range
transport of fungal spores from megacities (Beijing) by southwest winds govern the diurnal
variation of sugar alcohols in the Mangshan atmosphere. On the other hand, lower concentrations
in nighttime can be explained by the clean air mass transport by mountain breeze from the
Mangshan National Forest area.
**3.2. Source apportionment of SCs**

To investigate the source apportionment of SCs, positive matrix factorization (PMF)

software version 5.0 (Environmental Protection Agency, USA) was used. The PMF analysis was
performed for the measured aerosol samples using tracer compounds for anhydrosugars, primary
sugars, and sugar alcohols. It is essential to select a suitable number of factor solutions in the PMF
analysis. Based on the possible sources of SCs, four to six factor solutions were run in PMF
model. In the four-factor solutions, the SCs, including arabitol, mannitol, and trehalose, were
merged in a single factor; this might underestimate the soil dust sources. In six factor solutions,
the SCs, including glucose, fructose trehalose, arabitol, and mannitol, were distributed in more
than four factors; it might be overestimated the number of factor solutions according to possible
sources of SCs. Therefore, a total of five interpretable factor solutions were characterized by the
enrichment of each tracer compound to be significant to categorize the origins of individual
sugars, which reproduced more than 95% of SCs.

These five-factor solutions were preferred based on minimum robust and true Q values

(goodness of fit parameters) of the base runs, which observed 3103 and 3505, respectively. In each
bootstrap run, the concentrations and percentages of tracers were close to those of base-run results.
The PMF results of SCs indicate a stability because no significant changes were found between Q
values and factor profiles of F$_{peak}$ rotation runs compared with the base run. PMF results show a
good correlation between the values of observed and predicted (modeled) concentrations in scatter
plot, indicating that the model very well fits the individual sugar species. These results support the
perfect rationality of the source apportionment (Figure S-1). The time series plot of observed and
predicted concentration (modeled) also shown that the model well fits the observed data set
(Figure S-2). The time series plots of the factors solutions determined by PMF were similar to the
temporal plots of the concentration of sugar species of the factor composition (Figure S-3). The
numbers of factors were reduced if the pair of factors was strongly correlated. The composition of
each factor was also checked; none of the pair of factors were found with similar composition. We
also investigated the change in factor profile with positive and negative values of $f_{peak}$ for the
chosen solution in the PMF analysis. Figures 6 and 7 show the factor profile resolved by PMF
analysis of the Mangshan aerosol samples. The percentages of each component are summed for
factors 1 to 5 to be calculated as 100%.
Factor 1 is characterized by the high contribution of glucose (80.2%) followed by fructose
(69.6%), mannitol (24.8%), and inositol (15.1%) (Fig. 7a). Glucose and fructose are highly water-
soluble SCs present in the leaves and bark of plants (Graham et al., 2003). High concentrations of
glucose and fructose have been reported in vascular plants and phytoplankton by Cowie and
Hegdes (1984). The dominant glucose and fructose in the Mangshan aerosol samples collected in
autumn are rational as leaf senescence and decay results in both primary sugars being released into
the atmosphere during the fall season. We found an excellent correlation between glucose and
fructose (r = 0.94) in the Mangshan aerosols (Table 2), indicating the similar vegetation sources
for both sugar species in autumn (Baker et al., 1998; Burshtein et al., 2011; Pacini, 2000). Higher
concentrations of glucose and fructose in the aerosol samples collected during the autumn season
are reasonable because leaf senescence and decay result in an increased emission of primary
sugars into the atmosphere.
Several studies have reported that plant species significantly contribute to mannitol in the
atmosphere (Burshtein et al., 2011; Devis et al., 1988; 1990). Miyazaki et al. (2014) also found a
significant amount of trehalose, mannitol, and arabitol in the aerosol samples collected from the
forest and concluded their origin from the terrestrial plants within the forest. Significant positive
linear correlations of mannitol with fructose in daytime (r = 0.79) and nighttime (r = 0.86) further
denote that abundance of mannitol is due to the decay of plant leaves in autumn (Table 2).
Therefore, we conclude that the contributions of mannitol is from both vegetation and fungal
spores in the Mangshan aerosol samples. Hence mannitol showed the presence in factor 1.
Vegetations contribute to SCs during the campaign. Therefore, factor 1 can be termed as a
vegetation factor due to the high abundances of glucose, fructose, and mannitol.

Factor 2 is dominated by high loading of trehalose (80.2%), followed by mannitol (29.7%),
glucose (19.8%), and arabitol (18.2%) (Fig. 7b). The contribution of trehalose to soil dust has been
reported in several studies from different locations around the world, suggesting trehalose as a
tracer for the surface soil (Jia et al., 2010; Medeiros et al., 2006). In addition, previous studies
reported that bacteria and other microbes in the soil are also an essential source of trehalose
(Rogge et al., 2007). Trehalose is significantly correlated with arabitol (r=0.58) and mannitol
(r=0.58), and $Ca^{2+}$ (r=0.70), demonstrating its microbial and soil dust origin. Therefore, factor 2
can be termed as microbial and soil dust factor.

Factor 3 is characterized by levoglucosan (82.2%), galactosan (77%), and mannosan
(73.6%) (Fig. 7c). Previous studies have reported that these SCs are associated with BB aerosols
(Fraser and Lakshmanan, 2000; Graham et al., 2002; Simoneit, 2002). Simoneit et al. (1999)
reported that the pyrolysis of cellulose and hemicellulose emitted levoglucosan, galactosan and
mannosan. These sugar species are major organic components emits in the atmosphere by BB
activities (Simoneit et al., 2002). The BB influenced aerosols are enriched with levoglucosan,
mannosan, and galactosan (Nolte et al., 2001; Medeiros et al., 2006). The domestic BB for
cooking and house heating due to low ambient temperature and field burning of agricultural
residues occur in East Asia (Verma et al., 2015). The PMF results are very well supported by the
fact that anhydrosugars are associated with BB (Simoneit et al., 1999). Therefore, factor 3 can be
termed as a BB factor due to the high abundance of BB products.

Factor 4 is dominated by high loading of sucrose (90%), followed by inositol (36.9%) and
fructose (11.7%) (Fig. 7d). Sucrose plays a crucial role in the plant blossoming process as the
dominant sugar compound of pollen grains (Pacini, 2000). Several studies also reported that
sucrose is abundant sugar species found in airborne pollen grains and flowering plants (Fu et al.,
2012; Graham et al., 2003; Medeiros et al., 2006; Pacini, 2000). Therefore, sucrose is reported as
an excellent tracer for airborne pollen spores (Pacini, 2000). Thus factor 4 is termed as pollen
factor due to the high loading of sucrose.

Factor 5 is characterized by a higher contribution of arabitol (61.5%) followed by mannitol
(39.3%) and inositol (15.3%) (Fig. 7e). Sugar species contributing to factor 5 are associated with
fungal spores (Bauer et al., 2008). Various fungi and microbes emit spores, which are tracers for
the arabitol and mannitol; therefore, both sugars are considered as specific tracers of fungal
activities (Medeiros et al., 2006; Rogge et al., 2007). Thus, factor 5 is termed as a fungal factor
due to the high loading of arabitol and mannitol. Overall, the average contributions of each factor
to measured SCs were estimated by PMF analyses (Fig. 8), in which BB was found to account for
27% of measured SCs. The vegetation and microbial and soil dust sources equally contribute
(21%) to total SCs. The fungal spores and pollen spores contribute 16% and 15% of total SCs,
respectively. Finally, biomass burning emissions from the local areas and megacities via long-
range atmospheric transport were identified as an important source for the Mangshan aerosols.
**3.3. Contributions of sugar compounds to WSOC and OC**

The contribution of carbon content of measured SCs varied from 14.1-371 ng m$^{-3}$ (av.
145 ng m$^{-3}$) in daytime and 12.8-322 ng m$^{-3}$ (av. 117 ng m$^{-3}$) in nighttime, accounting for 0.83%
and 0.91% of OC, respectively (Fig. 9a, b). The mean carbon contents of anhydrosugars showed
clear diurnal variation with higher nighttime values (67.1 ng m$^{-3}$) than daytime (42.7 ng m$^{-3}$),
accounting for 0.43 % and 0.22 % of OC, respectively. These results suggest that BB significantly
contributed to Mangshan aerosols. However, the carbon contents of primary sugars showed
opposite diurnal variations; higher (68.5 ng m$^{-3}$) in daytime than nighttime (28.3 ng m$^{-3}$),
accounting for 0.41 % and 0.28 % of OC, respectively (Fig. 9a, b). This study suggests that the
daytime emissions of primary sugars from local vegetation and the decay of plant leaf in forest
significantly contribute to OC. The carbon concentration contributed by sugar alcohols showed
insignificant diurnal variations i.e. 34.6 ng m$^{-3}$ in daytime and 21.3 ng m$^{-3}$ in nighttime,
accounting for 0.20 % and 0.19 % of OC, respectively. This result indicates multiple carbon
sources of sugar alcohols in day and night. In addition, contributions of anhydrosugars, primary
sugars, and sugar alcohols to WSOC were similar to those of OC in Mangshan aerosols.

Based on the PMF analysis, we found five sources for SCs measured in Mangshan
aerosols. The different tracer compounds were used to calculate carbon contents: biomass burning-
C (i.e., levoglucosan, galactosan, mannosan), vegetation-C (glucose, fructose), fungal-C (arabitol,
mannitol), pollen-C (sucrose), and microbial-soil-C (trehalose) (Fig. 9c, d). Among the five
sources, biomass burning-C was found as the largest carbon contributor to Mangshan aerosols
(36.7%), followed by fungal-C (23.7%), vegetation-C (19.7%), pollen-C (14.2%), and microbial-
soil-C (4.84%). Biomass burning-C accounted for 1.38% and 0.43% at night, while 0.57% and
0.22% in daytime for WSOC and OC, respectively. The BB for cooking and space heating in
winter and autumn seasons are common in central China (Akagi et al., 2011), which should
increase the nighttime levels of Biomass burning-C at the Mangshan site. However, the carbon
contribution by vegetation and fungal sources are similar during day and nighttime for the
Mangshan aerosols. Pollen-C accounted for 0.20% and 0.07% of OC in daytime and nighttime,
respectively. Higher pollen activities are key sources for the high daytime levels of pollen-C in the
forest site (Taylor et al., 2002).

## 3.4. Contribution of levoglucosan to OC and WSOC

We calculated the mass concentration ratios of levoglucosan to OC (Lev/OC) and WSOC (Lev/WSOC) to evaluate the contributions of BB and anthropogenic emissions to Mangshan aerosols (Fig. 9a-c). Fossil fuel combustion and BB emit WSOC and OC. They are also secondarily produced by photochemical oxidation of volatile organic compounds in the atmosphere (Wang et al., 2005a; Deshmukh et al., 2019b). Coal combustion and vehicle exhaust can contribute to the high levels of OC and WSOC in aerosols (Xu et al., 2020). Levoglucosan, a dominant constituent of BB products, has been considered as an excellent tracer of BB (Simoneit, 2002; Kuo et al., 2011).

Average Lev/OC ratio ($5.69 \times 10^{-3}$) was lower than that of Lev/WSOC ($1.66 \times 10^{-2}$) in Mangshan samples (Fig. 10a). Yan et al. (2018) reported similar ratios of Lev/OC ($4.0 \times 10^{-3}$) and Lev/WSOC ($1.6 \times 10^{-2}$) for coal combustion, suggesting a significant carbon contribution to Mangshan aerosols from coal combustions in the industrial areas via long range transport. Interestingly, we found a substantial diurnal variation of Lev/OC and Lev/WSOC ratios. The average Lev/OC and Lev/WSOC ratios are several times higher in nighttime ($8.48 \times 10^{-3}$ and $2.70 \times 10^{-2}$, respectively) than daytime ($4.21 \times 10^{-3}$ and $1.11 \times 10^{-2}$, respectively) (Fig. 10b, c). These results indicate that BB contributed substantially to the Mangshan organic aerosols in nighttime. Moreover, the correlations of levoglucosan with OC and WSOC are stronger in nighttime (r = 0.81 and 0.70, respectively) than daytime (r = 0.45 and 0.40, respectively), demonstrating the dominance of BB-derived aerosols in the nighttime Mangshan samples (Table 2).

In addition, WSOCs are derived from various emission sources. We propose that secondary organic aerosols constitute a significant fraction of WSOC and OC in daytime Mangshan aerosols. The photochemical oxidation of organic precursors emitted from fossil fuel combustion in industries and vehicular exhausts also contributes to secondary production of WSOC and OC in daytime (He et al., 2015), suggesting that emissions from the urban Beijing area may significantly influence the daytime levels of Mangshan aerosols. He et al. (2015) proposed a possible contribution of photochemical formation of secondary organic aerosols to atmospheric WSOC and OC in north China. Nevertheless, the photochemical degradation of levoglucosan by OH radicals under ultraviolet radiations and high temperatures (Hennigan et al., 2010) may play a key role in lowering the ratios of Lev/OC and Lev/WSOC in daytime Mangshan aerosols.

## 3.5. Mass concentration ratios of levoglucosan/mannosan

The mass concentration ratios of levoglucosan and mannosan (Lev/Man) were calculated to better characterize the emissions sources of BB tracers (softwood vs. hardwood) in the

Mangshan site. Figure 10d represents the variations of Lev/Man ratios for overall, day- and
nighttime periods. The Lev/Man ratios have been used to distinguish the hardwood (angiosperm)
and softwood (gymnosperm) burning in the ice core record from the Russian Far East (Kawamura
et al., 2012). Hardwood contains 55–65% cellulose and 20–30% hemicellulose (Klemm et al.,
2005). Levoglucosan and mannosan are derived from the thermal decomposition of cellulose and
hemicelluloses, respectively (Simoneit, 2002). Levoglucosan is thermally more stable than
mannosan and galactosan (Kuo et al., 2011). Hence, a lower Lev/Man ratio is associated with
softwood burning, whereas a higher ratio is associated with hardwood burning (Engling et al.,
2006, 2009). However, we found insignificant diurnal variations of Lev/Man ratios between night
(9.33-25.9, avg. 15.8) and daytime aerosols (0.90-23.3, avg. 13.6). Likewise, comparable Lev/Man
ratios (9-13 for $PM_{10}$ and 10-13 for $PM_{2.5}$) were reported for aerosol samples from Tanzania,
where wood and charcoal are primary fuels used for domestic cooking and heating (Mkoma and
Kawamura, 2013). Interestingly, wheat straws and lignite are used in China for domestic cooking
and house heating, which may also contribute to levoglucosan and mannosan in the Mangshan
aerosols.
Different Lev/Man ratios were reported in the chamber and controlled field experiments,
e.g., 4-22 for conifer and savanna grass burning (Iinuma et al., 2007), and 41.6 for rice straw and
and 55.7 cereal straw burning (Engling et al., 2009; Zhang et al., 2007). Kuo et al. (2011) reported
higher emissions of levoglucosan during high-temperature flaming (27.5-52.3) compared to low-
temperature smoldering (2.43-3.08). Hence, it is not easy to differentiate hardwood and softwood
burning based on Lev/Man ratios alone. Several studies reported a high Lev/Man ratio for both
softwood and hardwood burning. Thus, there may exist some other factors that significantly
control the Lev/Man ratios. Yan et al. (2018) found a significant contribution of levoglucosan in
coal combustion with Lev/Man ratio of 7.2. The variations of Lev/Man ratios in Mangshan may be
significantly influenced by several factors, i.e., flaming vs. smoldering, duration of biomass
burning, coal combustion, and hardwood vs. softwood burning. The moderate Lev/Man ratios in
autumn aerosols from Mangshan suggest that low temperature smoldering processes of hardwood
contribute to levels of levoglucosan and mannosan. However, the contribution of coal
combustions for house heating could not be excluded.
**4. Summary and Conclusions**
Anhydrosugars, primary sugars, and sugar alcohols were detected with distinct diurnal
variations in suburban aerosol samples collected at the Mangshan site in the northern vicinity of
Beijing. The wind patterns indicate that daytime air masses were transported from urban Beijing to
Mangshan, while clean air masses were delivered in nighttime from the Mangshan National Forest
Park. Daytime air masses from urban Beijing significantly influence the air quality of the northern
forest region. We observed the highest abundance of primary sugars, followed by anhydrosugars
and sugar alcohols. Local emissions from the forest plants and fungal species are the main
contributors to the primary sugars and sugar alcohols in the Mangshan aerosols. The
meteorological parameter significantly influenced the levels of SCs in the Mangshan samples. We
observed a significant influence of enhanced ambient temperature and solar radiation on the pollen
rupture and increased RH on fungal and microbial growth. This study suggested the source
variation for trehalose, i.e., local microbes at night and soil dust particles transported from Beijing
areas by southerly wind in daytime. We found that vegetation and fungal spores are not a specific
source of glucose and mannitol, respectively. Both sugars may have multiple sources in the forest
aerosols.
PMF results concluded the contributions of 36% from vegetation (21% vegetation factor
and 15% pollen factor) and 37% from microbial and fungal species (21% microbial soil dust and
16% fungal factor) of total measured SCs. The BB activities for domestic cooking and space
heating in north China contributed higher organic carbon at nighttime (0.43%) than in daytime
(0.22%). Therefore, local BB seriously affected the air quality of the Mangshan site. Lev/Man
ratio suggested that low temperature smoldering burning of hardwood is the main source for BB
aerosols. SCs were recognized as a significant aerosol component at Mangshan, northern suburbs
of Beijing. SCs can influence the air quality and thus climate because they are essential
components of organic aerosols on a global scale. This study of SCs at Mangshan demonstrates
that ambient levels of SCs are highly sensitive to the emissions of anthropogenic and biogenic
aerosols. Higher contribution of levoglucosan to SCs demonstrated a significant BB activity
around the Mangshan site in north China.

*Data availability.* Raw data are available on request by contacting the corresponding author.
*Author contributions.* This research was designed YK, KK and ZW. Laboratory measurements
were performed by FY with a support of PF. The paper was prepared by SKV and KK.
*Competing interests.* The authors declare that they have no conflict of interest.
*Financial support.* This research was supported by the Japan Society for the Promotion of
Science (JSPS) through grant-in-aid Nos. 19204055 and 24221001.
*Acknowledgements.* The authors thank Kazuhiko Okuzawa for his help during the sample
collection. This study was partly supported by the National Natural Science Foundation of China
through grant-in-aid No. 41625014. The financial support by the Global Environment Research
Fund (B-051) of the Ministry of the Environment, Japan should also be acknowledged for the
shipping of the sampling equipment to the Mangshan site.

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

**Figure Captions**

Figure 1. Geographical location of Mangshan, China. The map was downloaded from © Google Maps 2019.

Figure 2. (a) The meteorological parameters at Mangshan during sampling periods, (b) Fractions of local wind directions at Mangshan site, north of Beijing, China.

Figure 3. Concentrations (ng m$^{-3}$) of sugar compound (a) overall, (b) daytime and (c) nighttime in aerosol samples from Mangshan during September-October 2007 (The error bars denote the standard deviation).

Figure. 4. Temporal variations in the concentrations (ng m$^{-3}$) of anhydrosugars in the Mangshan aerosol samples collected for September-October 2007. (Solid circle represents nighttime samples collected from 18:00 to 09:00 hours. Hollow circle represents daytime samples).

Figure. 5. Temporal variations in the concentrations (ng m$^{-3}$) of primary sugars in the Mangshan aerosol samples collected for September-October 2007. (Solid circle represents nighttime samples collected from 18:00 to 09:00 hours. Hollow circle represents daytime samples). Y-axis shows temporal variations in the concentrations (μg m$^{-3}$) of Ca$^{2+}$.

Figure. 6. Temporal variations in the concentrations (ng m$^{-3}$) of sugar alcohols in the Mangshan aerosol samples collected for September-October 2007. (Solid circle represents nighttime samples collected from 18:00 to 09:00 hours. Hollow circle represents daytime samples).

Figure 7. PMF analyses of sugar compounds in Mangshan aerosols based on the autumn 2007 data set.

Figure 8. Source contributions to sugar compounds from various sources based on PMF analyses.

Figure 9. The concentrations and relative contributions of the carbon content of anhydrosugars, primary sugars and sugar alcohols to the carbon concentrations of measured sugar compounds, water-soluble organic carbon (WSOC) and organic carbon (OC) fraction of Mangshan aerosols (a = daytime and b = nighttime). The concentrations and relative contribution of the carbon content of five sources of sugar compounds to total sugar compounds measured, WSOC and OC fraction of Mangshan aerosols (c = daytime and d = nighttime).

Figure 10. Mass concentrations ratio of carbon contents of (a) levoglucosan (Lev) to organic carbon (OC) and water soluble organic carbon (WSOC), (b) levoglucosan (Lev) to organic carbon (OC) daytime and night time, (c) levoglucosan (Lev) to water soluble organic carbon (WSOC) daytime and night time, (d) average levoglucosan to mannosan ratios (Lev/Man) in the Mangshan aerosol samples for autumn 2007.



Fig. 1.




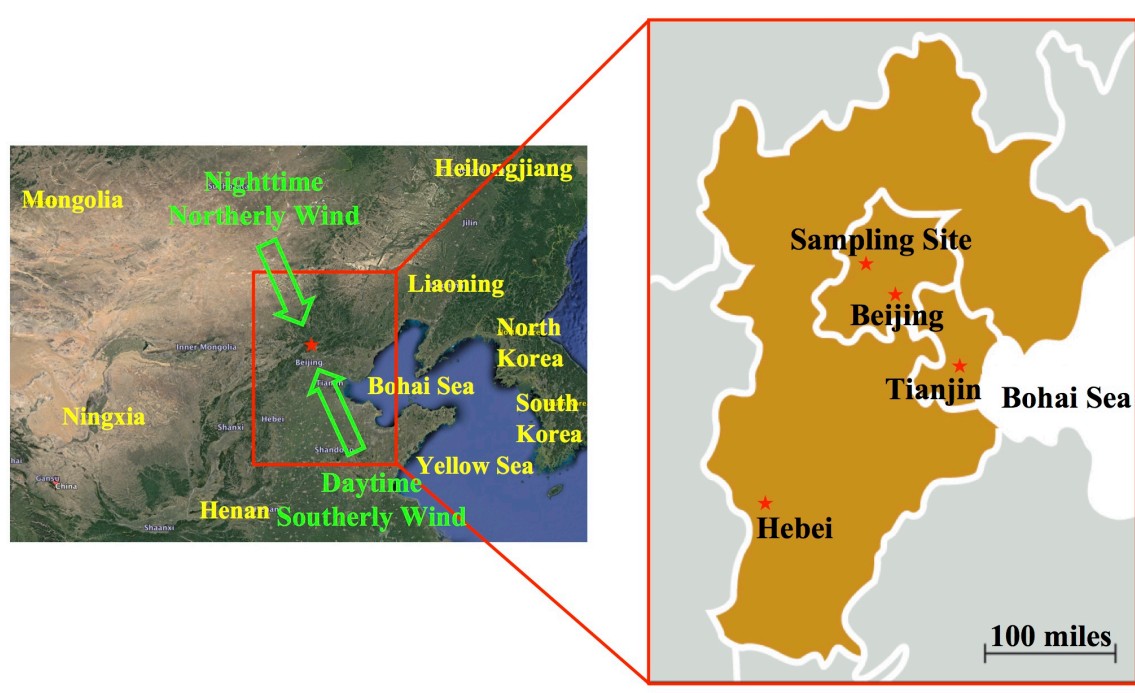














Fig. 2.



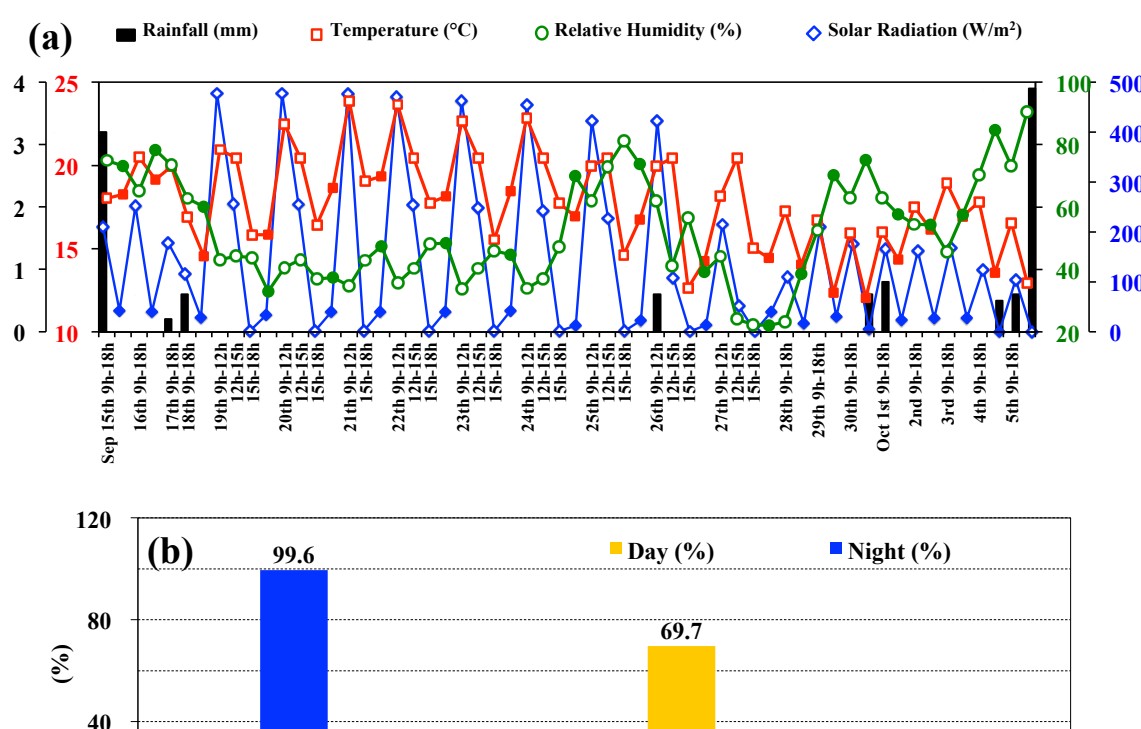






Fig. 3.



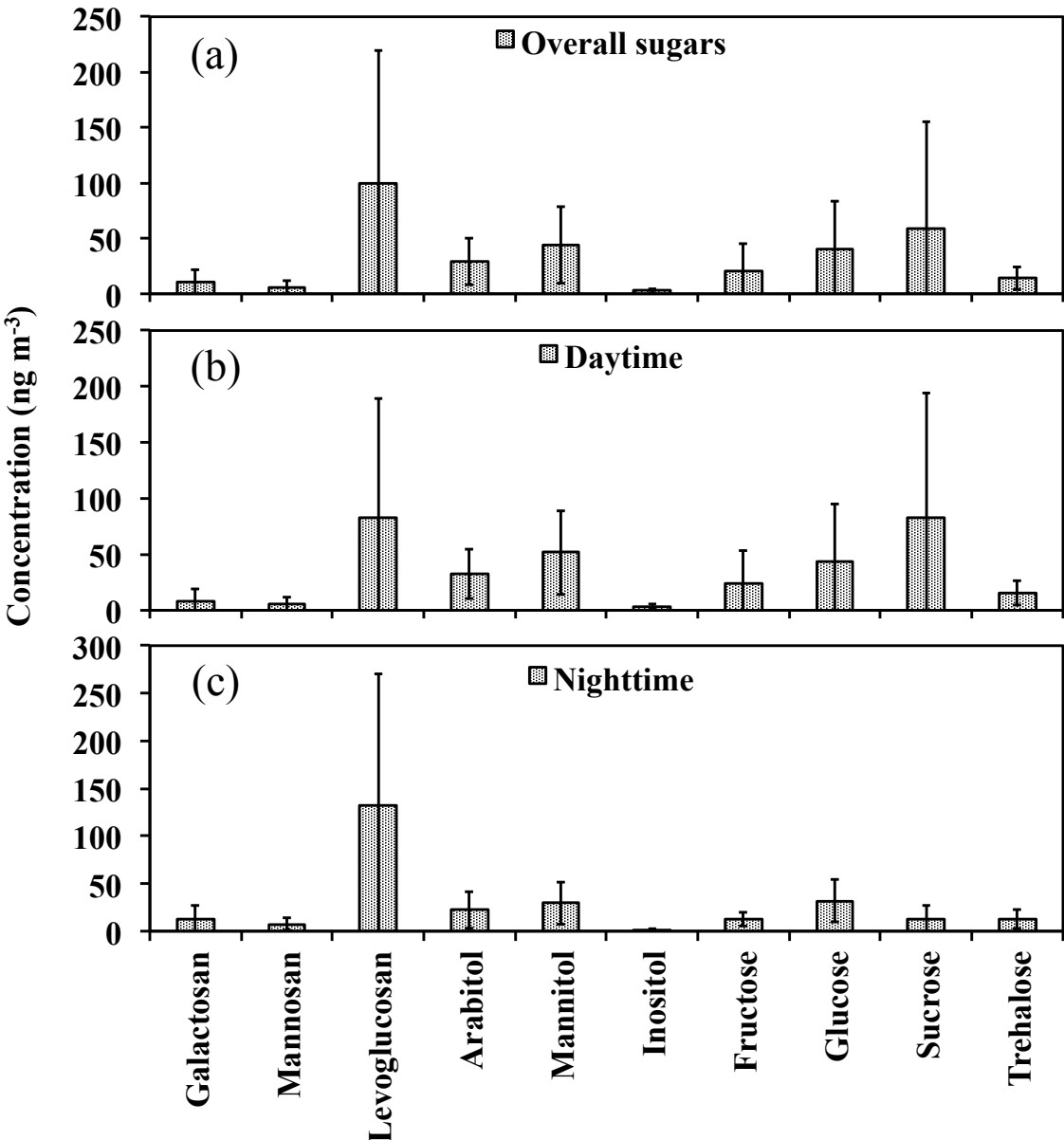








Fig. 4



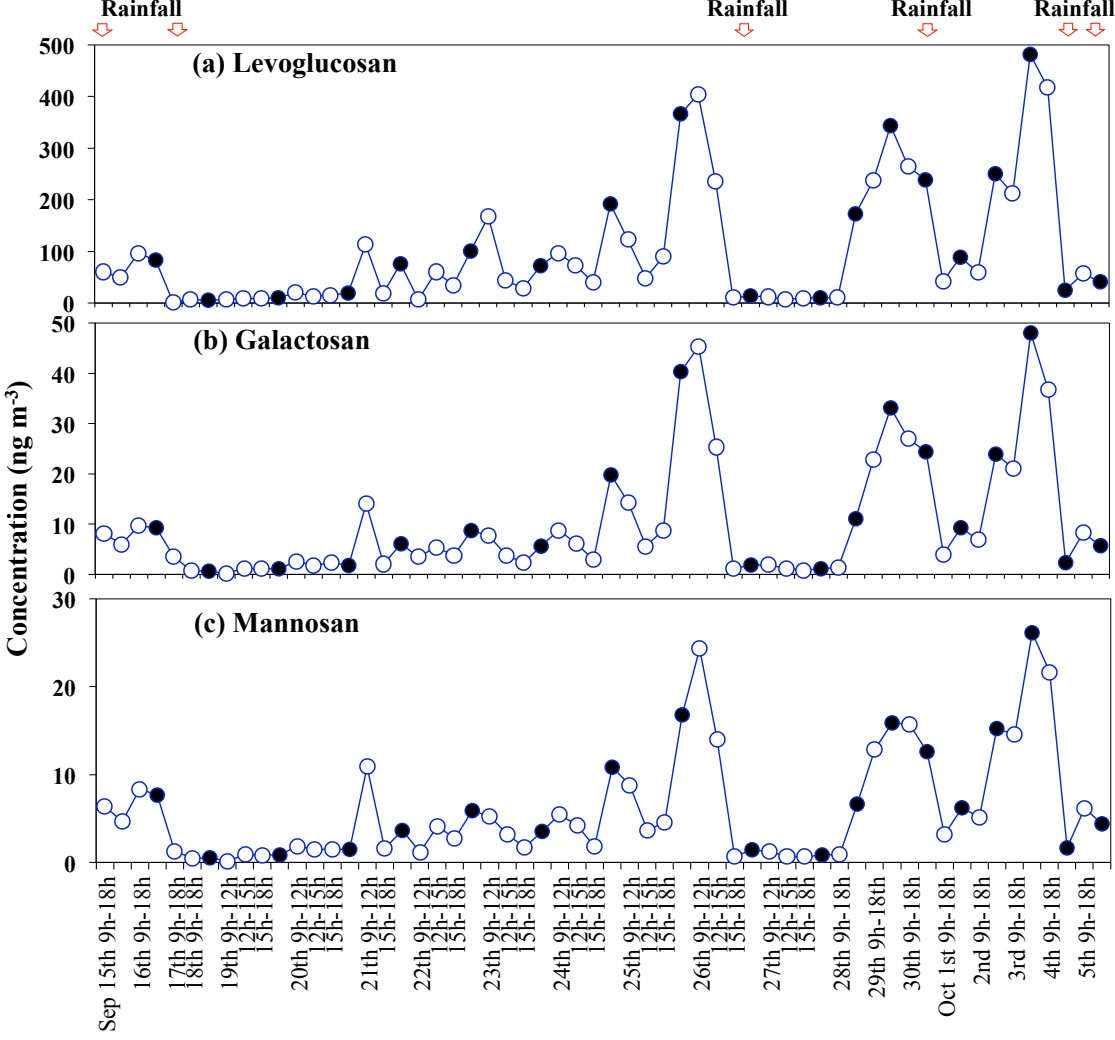











Fig. 5

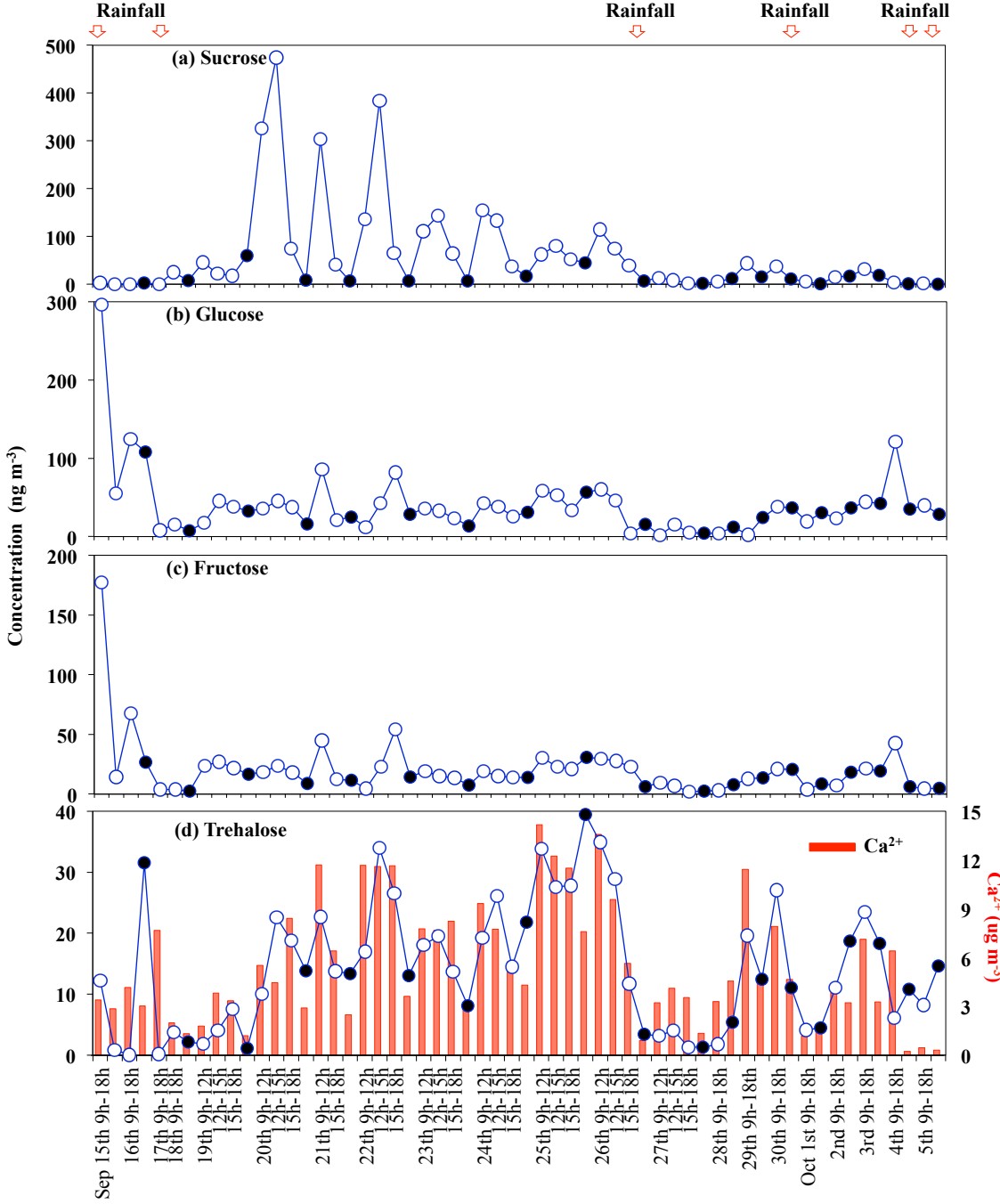









Fig. 6.


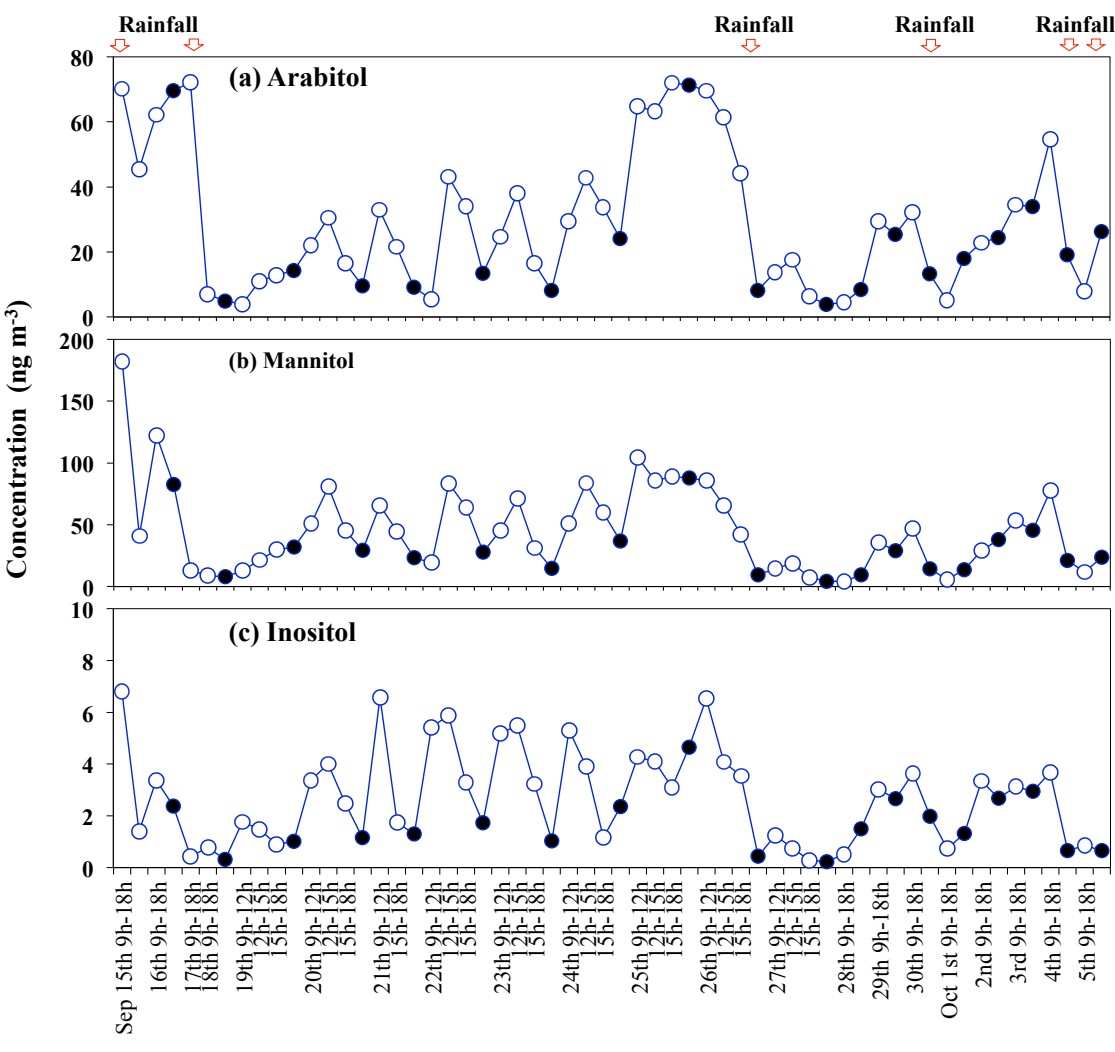







Fig. 7.

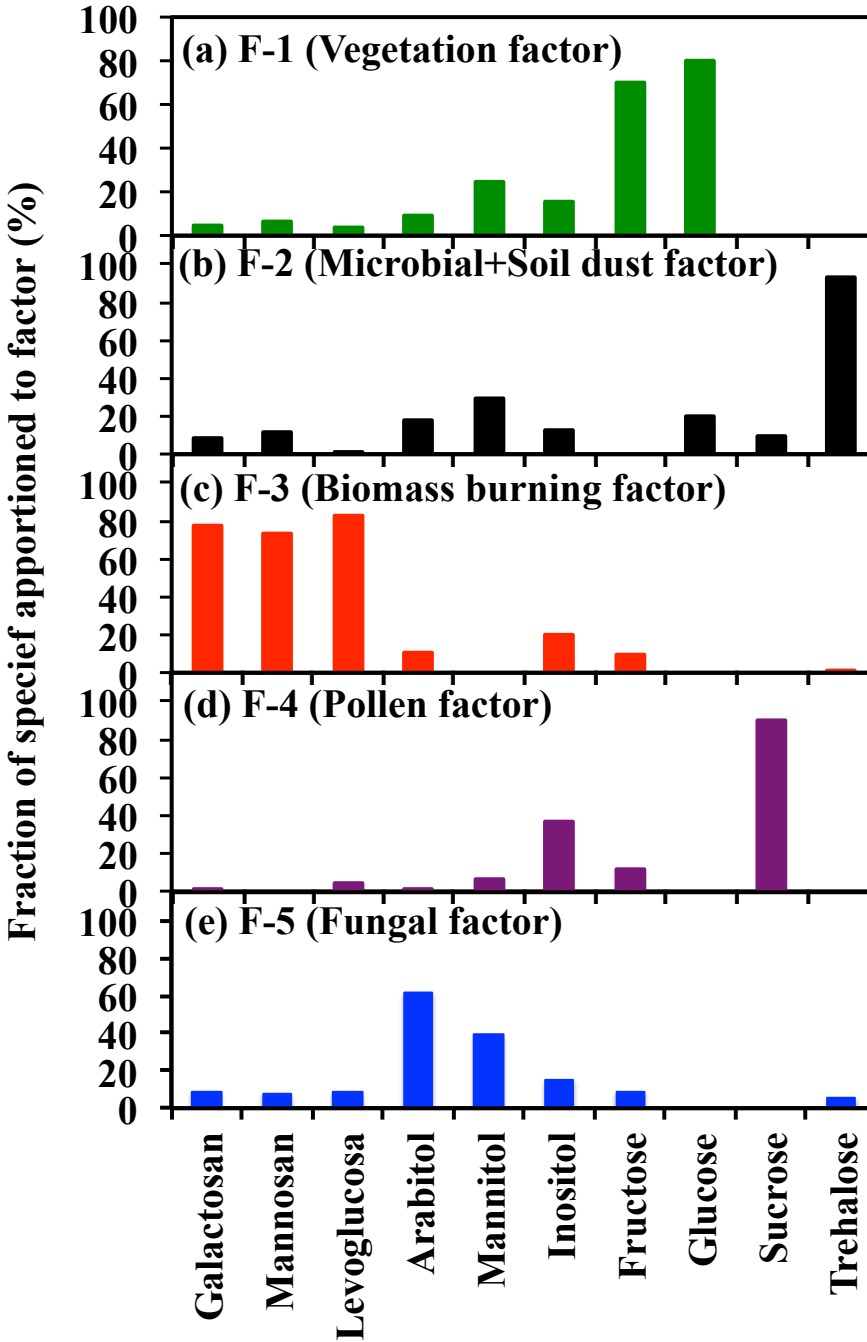




Fig. 8.



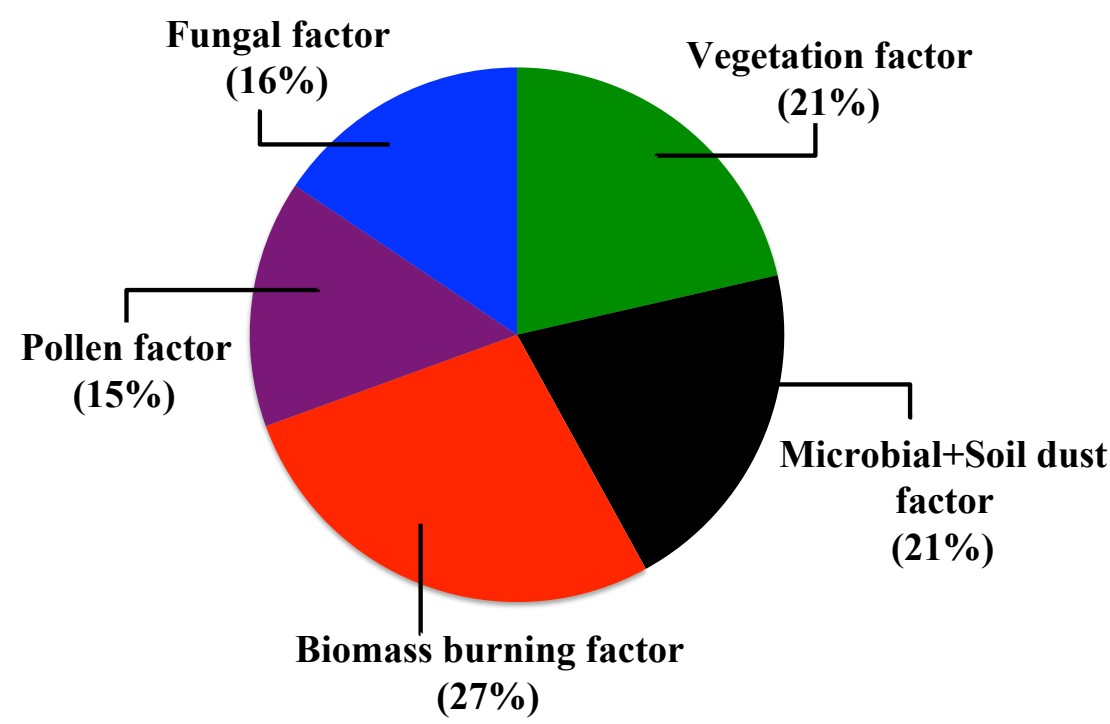

Fungal factor (16%)

Vegetation factor (21%)

Pollen factor (15%)

Microbial+Soil dust factor (21%)

Biomass burning factor (27%)




Fig. 9.

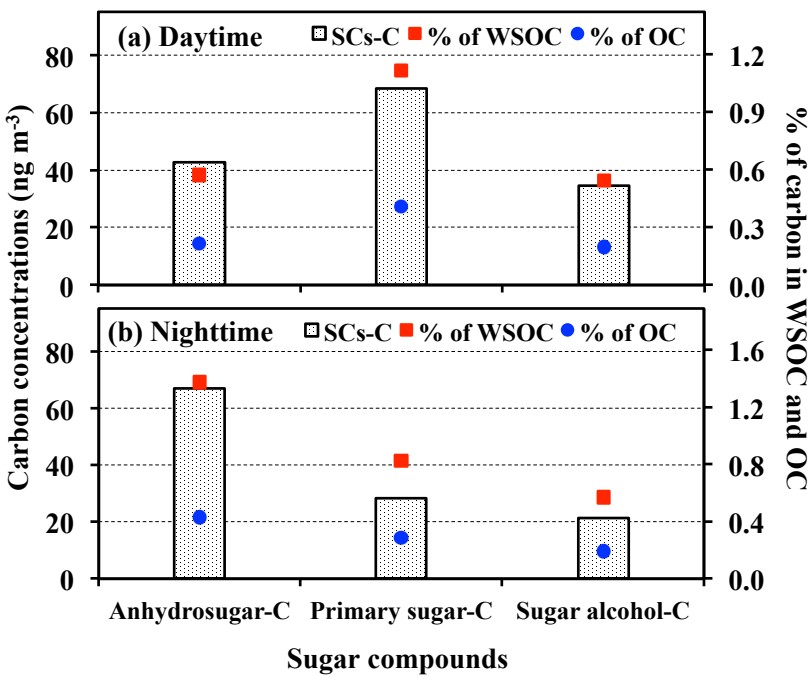

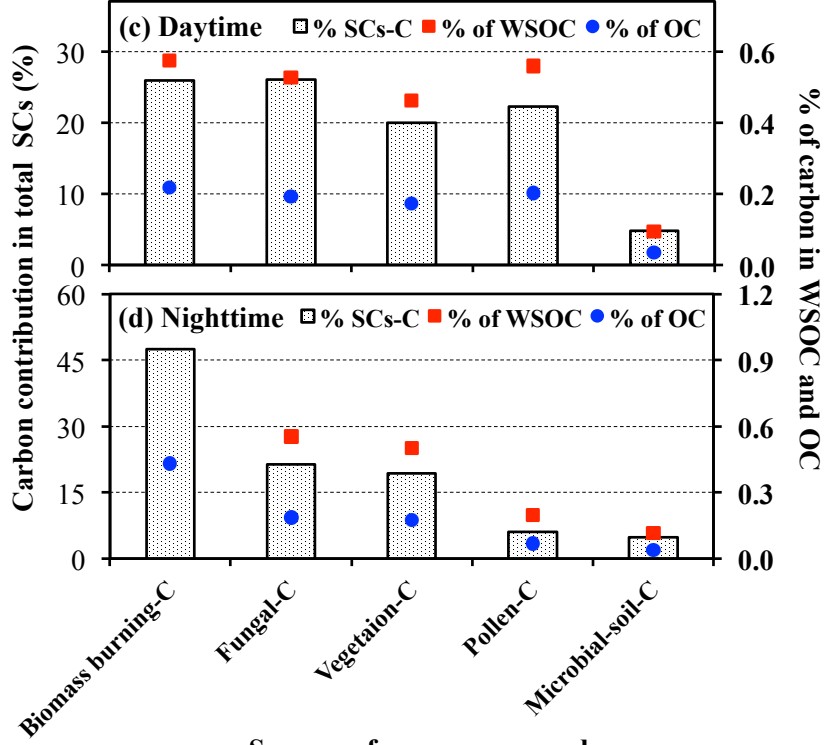


Fig. 10.

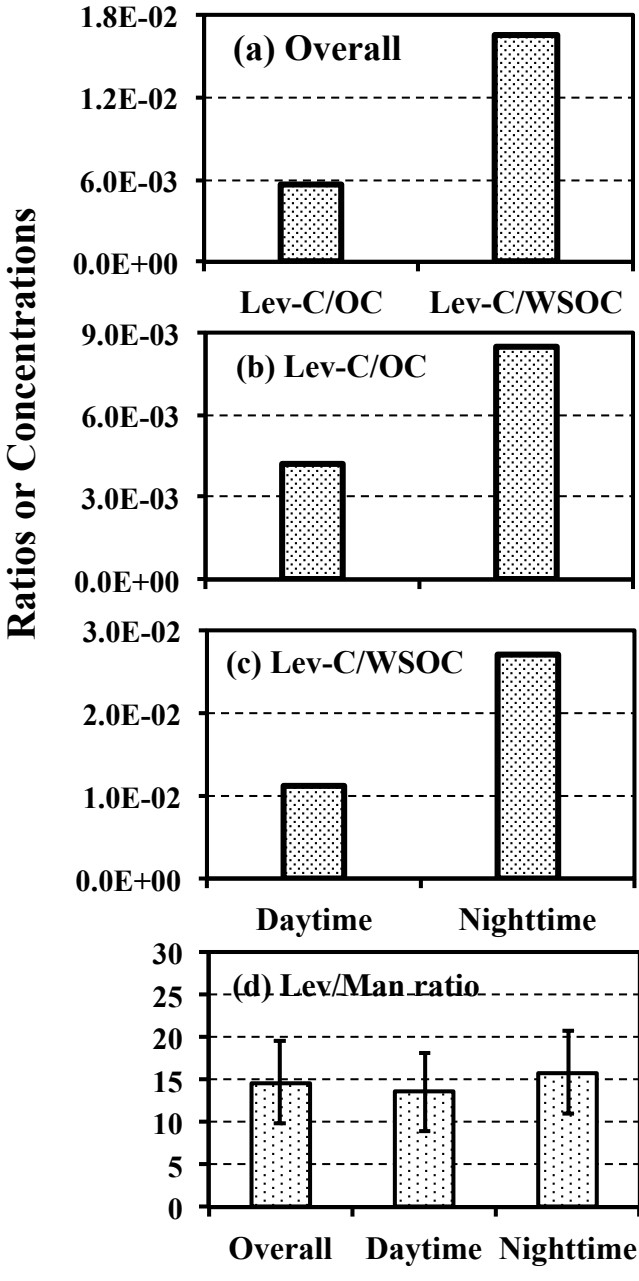


Table 1. Minimum, maximum, average and standard deviations of concentrations of sugar
compounds in aerosol samples (TSP) from Mangshan, China.

| Sugar Compounds | Overall | | | | Daytime (n = 38) | | | | Nighttime (n = 20) | | | |
|---|---|---|---|---|---|---|---|---|---|---|---|---|
| | Min | Max | Avg. | S.D. | Min | Max | Avg. | S.D. | Min | Max | Avg. | S.D. |
| **Anhydrosugars** | | | | | | | | | | | | |
| Galactosan | 0.14 | 48.0 | 10.1 | 11.9 | 0.14 | 45.3 | 8.53 | 10.5 | 0.69 | 48.0 | 13.0 | 14.0 |
| Mannosan | 0.13 | 26.1 | 6.05 | 6.33 | 0.13 | 24.3 | 5.37 | 6.01 | 0.53 | 26.1 | 7.35 | 6.87 |
| Levoglucosan | 1.17 | 482 | 100 | 119 | 1.17 | 418 | 83.2 | 106 | 5.66 | 482 | 132 | 138 |
| **Sugar alcohols** | | | | | | | | | | | | |
| Arabitol | 3.89 | 72.2 | 29.1 | 21.5 | 3.99 | 72.2 | 32.5 | 22.0 | 3.89 | 71.3 | 22.5 | 19.4 |
| Mannitol | 4.19 | 182 | 44.1 | 34.5 | 4.19 | 182 | 51.7 | 37.5 | 4.40 | 87.7 | 29.6 | 22.3 |
| Inositol | 0.23 | 6.8 | 2.62 | 1.81 | 0.27 | 6.80 | 3.14 | 1.90 | 0.23 | 4.65 | 1.62 | 1.09 |
| **Primary sugars** | | | | | | | | | | | | |
| Fructose | 1.72 | 177 | 20.1 | 24.6 | 1.72 | 177 | 23.9 | 29.3 | 2.64 | 30.9 | 12.8 | 7.67 |
| Glucose | 1.86 | 297 | 40.0 | 43.4 | 1.86 | 297 | 44.2 | 50.8 | 4.52 | 108 | 32.0 | 22.8 |
| Sucrose | 0.02 | 474 | 58.5 | 96.5 | 0.02 | 474 | 82.9 | 112 | 0.04 | 60.1 | 12.3 | 15.1 |
| Trehalose | 0.06 | 39.5 | 14.3 | 10.5 | 0.06 | 34.9 | 15.3 | 10.6 | 0.87 | 39.5 | 12.3 | 10.2 |
| Anhydrosugars | 6.01 | 556 | 116 | 137 | 6.01 | 476 | 97.1 | 122 | 6.88 | 556 | 152 | 159 |
| Primary sugars | 9.41 | 565 | 133 | 125 | 9.41 | 565 | 166 | 141 | 10.5 | 172 | 69.4 | 43.0 |
| Sugar alcohols | 8.53 | 259 | 75.8 | 54.7 | 9.09 | 259 | 87.4 | 57.5 | 8.53 | 164 | 53.7 | 41.9 |
| Total Sugars | 30.8 | 875 | 325 | 232 | 34.1 | 875 | 351 | 240 | 30.8 | 759 | 276 | 212 |
| Anhydrosugars (%) | | | 31.9 | | | | 24.6 | | | | 45.7 | |
| Primary sugars (%) | | | 41.8 | | | | 47.3 | | | | 31.3 | |
| Sugar alcohols (%) | | | 26.4 | | | | 28.1 | | | | 23.0 | |


Table 2. Statistical summary of correlations among the chemical species and meteorological variables in aerosol samples collected at a forest site in northern Japan

| Linear regression | Correlation coefficient | p value | Significance of correlation at P value < 0.05 |
|---|---|---|---|
| Overall (n = 58) | | | |
| Levoglucosan vs. Galactosan | 0.98 | < 0.05 | Significant |
| Levoglucosan vs. Mannosan | 0.97 | < 0.05 | Significant |
| Mannosan vs. Galactosan | 0.98 | < 0.05 | Significant |
| Sucrose vs. Temperature | 0.52 | < 0.05 | Significant |
| Sucrose vs. Solar radiation | 0.55 | < 0.05 | Significant |
| Arabitol vs. Mannitol | 0.81 | < 0.05 | Significant |
| Arabitol vs. RH | 0.69 | < 0.05 | Significant |
| Mannitol vs. RH | 0.57 | < 0.05 | Significant |
| Glucose vs. Fructose | 0.94 | < 0.05 | Significant |
| Trehalose vs. Arabitol | 0.58 | < 0.05 | Significant |
| Trehalose vs. Mannitol | 0.58 | < 0.05 | Significant |
| Trehalose vs. $Ca^{2+}$ | 0.70 | < 0.05 | Significant |
| Daytime (n = 38) | | | |
| Sucrose vs. $Ca^{2+}$ | 0.32 | > 0.05 | Not significant |
| Glucose vs. $Ca^{2+}$ | 0.02 | > 0.05 | Not significant |
| Trehalose vs. Arabitol | 0.49 | < 0.05 | Significant |
| Trehalose vs. Mannitol | 0.51 | < 0.05 | Significant |
| Trehalose vs. $Ca^{2+}$ | 0.81 | < 0.05 | Significant |
| Fructose vs. Mannitol | 0.79 | < 0.05 | Significant |
| Levoglucosan vs. OC | 0.45 | < 0.05 | Significant |
| Levoglucosan vs. WSOC | 0.40 | < 0.05 | Significant |
| Nighttime (n = 20) | | | |
| Sucrose vs. $Ca^{2+}$ | 0.37 | > 0.05 | Not significant |
| Glucose vs. $Ca^{2+}$ | 0.27 | > 0.05 | Not significant |
| Trehalose vs. Arabitol | 0.76 | < 0.05 | Significant |
| Trehalose vs. Mannitol | 0.85 | < 0.05 | Significant |
| Trehalose vs. $Ca^{2+}$ | 0.61 | < 0.05 | Significant |
| Fructose vs. Mannitol | 0.86 | < 0.05 | Significant |
| Levoglucosan vs. OC | 0.81 | < 0.05 | Significant |
| Levoglucosan vs. WSOC | 0.70 | < 0.05 | Significant |

The data of $Ca^{2+}$, OC and WSOC are adapted from He et al. (2015).