# Peer review of "Measurement report: Diurnal and temporal variations of sugar compounds in suburban aerosols from the northern vicinity of Beijing, China: An influence of biogenic and anthropogenic sources"

_Atmospheric Chemistry and Physics, 2020_

## Referee Comment (RC1) · Anonymous Referee #1 · 20 Oct 2020

Comments on "Measurement report: Diurnal and temporal variations of sugar compounds in suburban aerosols from the northern vicinity of Beijing, China: An influence of biogenic and anthropogenic sources" by Verma et al., October 2020.

General Comments:

In this manuscript the authors report observations of sugar compounds (SCs) in air at a rural site about 40 km north of Beijing from Aug 15- Oct 5, 2007. Diurnal variability

is examined, and meteorological parameters are considered as explanatory variables. The SC time series were analyzed with positive matrix factorization to identify sugar aerosol types and the relative contributions of the sugars and aerosol types to organic aerosol mass are reported. Overall this is a useful contribution of measurements to a topic that is still not well-understood. The differentiation between daytime and nighttime samples is rightly recognized as important by the authors. However, the manuscript often reads like a laundry list of observations interspersed with comparisons to previous observations, and insufficient analysis to support some claims.

Some inferences made about the observations are provided as speculations with little explanation or supporting analysis. In fact there are a few claims made in the manuscript (noted below) that don't even seem to clearly follow from the evidence presented. The logic behind the claims needs to be clarified or the claims need to be changed or removed. A major driver of this issue is that day-night differences were also differences in temperature and humidity, and differences in air mass origin. How can these effects be disentangled to draw inferences? For example, sucrose is interpreted to be controlled by local emission related to temperature and radiation on the basis of correlation with those variables, and transport is not considered. But arabitol and mannitol correlate very closely with local RH- why in the Abstract is it claimed that these are related to transport from Beijing? Perhaps this would be clearer if a more comprehensive table of correlation coefficients were shown for the relevant quantities, for daytime, nighttime, and overall, but some more textual clarification would also help. Also, would be really nice to have some proxy of transport there too, e.g. average magnitude of the wind in the direction between Beijing and the site.

A major limitation is the lack of any air mass trajectory analysis. The authors state that there was a typical diurnal pattern to the wind direction, with daytime winds from the south and the large cities, and nighttime winds from more rural areas to the north. It would help the reader to see how consistent this pattern was in order to evaluate some of the claims made. I suggest at least a time series plot of wind speed and

wind direction, perhaps as a sub-plot to Figure 2, or a wind rose diagram. A couple of representative air mass back trajectories could be instructive as well.

The results section should have more description of the present data set, with more of the previous observations moved to the Introduction. In particular, the description of the PMF results could be expanded, perhaps with a figure showing the time series of the PMF factors.

Throughout the manuscript, the authors should to be more careful in their descriptions of how the SCs get into the atmosphere. They frequently state that a process or an organism "emits" a sugar compound, which reads somewhat ambiguously (i.e. are gas phase compounds being released?). I think it's OK to use this language, but there first needs to be a clear statement in the Introduction about how SCs get into the atmosphere, or at least the state of the science on that question. How are SCs are released into the air, as fragments of organisms, as whole fungal spores, as individual molecules, etc?

A further issue that needs to be resolved before this is publishable is the extremely sparse description of the methods. Specific questions are raised below.

At this point this manuscript is essentially a descriptive account of measurements made at a particular location, with some interpretive claims that seem a bit ambiguous. The measurements themselves are of value, but I think for publication, it needs 1) a much more thorough method description section and 2) either a) a scale-back of the claims made, or b) additional analysis in support of the claims.

Specific comments:

There are several grammatical issues of subject-verb agreement and lack of pluralization throughout the manuscript. They don't usually impede understanding, but the manuscript would benefit from a thorough grammar check.

It would be very helpful to view the data in Figure 2 directly as a part of Figure 3.

Line 66: "SCs are emitted from algae, microbes, pollen, suspended soil particle[s], and associated biota into the atmosphere" This statement reads to me a bit like sugar compounds might be released into the air as individual gas phase molecules, which I don't think is the case (?). Maybe it could be phrased "SCs are emitted as part of aerosols formed from algae..."? Same thing at line 72: I don't think mannitol and arabitol are mostly emitted as individual gases, but are a part of fungal material that gets into the atmosphere. What is physically meant here needs to be a little clearer. Are these SCs usually part of biological fragments, or is this unknown?

Section 2.2:

Please provide more details of the sampling apparatus and methodology. Were these samples collected with a high-volume sampler? Where was it installed specifically, and at what height? What aerosol sizes were collected? Were there any measures to avoid the sampling of gas phase components? Did the 3-hr and 9-hr samples overlap in time? What times were the samples started? Perhaps a table of sample collection times would be helpful.

Furthermore, what were the methods for determining the WSOC and total OC? How was Ca(2+) concentration determined? Was the filter cut into sections for each analysis?

Line 132: What is meant by C13 n-alkane? Is this an isotope standard of one n-alkane? Which one? Or do you mean C13H28, n-tridecane?

Line 157: "Hence, it is evident that increased BB activities at nighttime are associated with cool temperature (Fig. 2)." Is this saying that because it's cool at night, it makes sense that there's more BB aerosol at night? Isn't it equally likely that the different air mass origins in the day and at night are the reason?

Line 209: "the meteorological conditions". Is this referring to the strong daytime winds and convective activity? It would be clearer to state that directly.

Line 254: "northeasterly (99.5%)". Does this mean 99.5% of the nighttime hours the wind was northeasterly? Please clarify in the text.

Line 255: What would cause sugar emissions to decrease with lower temperature? Is there supporting literature for this?

Line 268: Trehalose paragraph. Trehalose didn't show a strong diurnal cycle, but the authors point out a correlation between trehalose and mannitol and arabitol at night, and between trehalose and Ca(2+) in the day. It would be helpful to at least report the corresponding correlation coefficients for the day and night, respectively, for comparison, and possibly to include the corresponding figures in Figure 5.

Line 272: Why would nighttime low RH and temp cause microorganisms to emit more trehalose? Please cite a reference. Again, the use of "emit" here can be confusing. Is it the release of spores that prefers these conditions?

Line 310: Aren't the Mt. Tai measurements higher than Mangshan, not lower?

Lines 315-319: I don't understand the reasoning here. How does RH relate to transport from megacities as an explanation for fungal aerosol?

Line 350: Separate the Factor 3 and Factor 4 descriptions into separate paragraphs.

Line 352: "The PMF results are very well supported by the fact that anhydrosugars are associated with BB in the Mangshan site." Is this referring to results from a previous study? Please cite it.

Line 410: "Our results also denote that secondary production of OC and WSOC from BB-derived organic precursors was crucial during nighttime at the Mangshan site." What evidence shows this? And do you mean that organic compounds went through chemical changes to form aerosol OC and WSOC, or simply that organics produced during biomass burning were incorporated into aerosol after the burning? In either case, I don't see how we know that.

---

## Referee Comment (RC2) · Anonymous Referee #2 · 21 Oct 2020

Verma et al. discuss observations of sugars found in the aerosol-phase collected for ~1 month in a forested site north of Beijing. The aerosol were collected onto filters and analyzed for the sugars. The authors then describe the pattern of the various sugars throughout the study period and speculate the sources via differences in day- and night-time mass concentration, wind patterns, and PMF. They discuss 5 potential sources, including biomass burning, vegetation, microbial and soil dust, pollen, and fungal.

[Figure]

The results presented here may be of interest to the audience and its scope generally fits within a measurement report. However, along with the concerns discussed by Reviewer #1, the authors nee to address the comments and concerns presented below prior to consideration for publication in ACP.

Major: (1) Statistics: Throughout the text, the authors state that the results are statistically different. However, conducting the t-test with the mean and standard deviation values listed in the table, majority of the observations are statistically similar at the 95% confidence interval and not statistically different. The lack of statistical difference in the observations makes many of the statements the authors use to differentiate day/night and thus sources less substantiated. Further, the correlations shown by the authors in Fig. 5 have very low R values (as stated throughout the text) and suggest that many of the correlations only explain 50% or less of the mass concentration.

(2) Contextualization of results: I agree with Reviewer #1 that the listing of numbers from prior results makes it difficult to understand the conclusions in each section and the whole paper. Further, as highlighted with point (1) above, the data not being statistically different makes sections 3.1.1 thru 3.1.3 very long and repetitive. Also, the listing of numbers from prior studies to ascribe sources for the sugars makes the source apportionment very uncertain. This is also relevant for Section 3.5, where they found no differences in the levoglusoan/mannosan ratio and spend 1.5 pages on this. If this is important, it could be summarized in one paragraph at most.

(3) Methods: Reviewer #1 highlighted many of the methods that should be discussed in more detail. Further, PMF needs to be described in more detail to understand how the 5 results were determined (e.g., how many solutions were there allowed to be, how did the time series look, were the results compared against and investigated against external variables, etc.). Also, agree with Reviewer #1 in how were WSOC, OC, Ca2+, etc determined.

(4) PMF: I think this is the more interesting and compelling part of the paper. I highly

[Figure]

**[ACPD](ACPD)**

Interactive
comment

recommend the authors spend more time expanding on this section while reducing the discussion in the other sections. As highlighted above, there are statistical concerns, thus shortening them while increasing the discussion about PMF, which had lower statistical concern.

(5) Figures: The x-axis/date is very hard to read in all figures. It is unclear what the values are shown in different colors in Fig. 3.

Minor: Please review the grammar throughout the paper, as highlighted by Reviewer #1.

---

## Author Response (AR1)

Comments on "Measurement report: Diurnal and temporal variations of sugar compounds in suburban aerosols from the northern vicinity of Beijing, China: An influence of biogenic and anthropogenic sources" by Verma et al., October 2020.

General Comments:

In this manuscript the authors report observations of sugar compounds (SCs) in air at a rural site about 40 km north of Beijing from Aug 15- Oct 5, 2007. Diurnal variability is examined, and meteorological parameters are considered as explanatory variables. The SC time series were analyzed with positive matrix factorization to identify sugar aerosol types and the relative contributions of the sugars and aerosol types to organic aerosol mass are reported. Overall this is a useful contribution of measurements to a topic that is still not well-understood. The differentiation between daytime and nighttime samples is rightly recognized as important by the authors. However, the manuscript often reads like a laundry list of observations interspersed with comparisons to previous observations, and insufficient analysis to support some claims.

Response: Authors are thankful to the reviewer for his valuable comments and suggestions, which help to upgrade the quality of the manuscript. We make significant changes in the manuscript especially in section 2. Materials and methods and 3.1. - 3.2. Results and discussion. We deleted several phrases of comparisons with previous observations.

Some inferences made about the observations are provided as speculations with little explanation or supporting analysis. In fact there are a few claims made in the manuscript (noted below) that don't even seem to clearly follow from the evidence presented. The logic behind the claims needs to be clarified or the claims need to be changed or removed. A major driver of this issue is that day-night differences were also differences in temperature and humidity, and differences in air mass origin. How can these effects be disentangled to draw inferences? For example, sucrose is interpreted to be controlled by local emission related to temperature and radiation on the basis of correlation with those variables, and transport is not considered. But arabitol and mannitol correlate very closely with local RH- why in the Abstract is it claimed that these are related to transport from Beijing? Perhaps this would be clearer if a more comprehensive table of correlation coefficients were shown for the relevant quantities, for daytime, nighttime, and overall, but some more textual clarification would also help. Also, would be really nice to have some proxy of transport there too, e.g. average magnitude of the wind in the direction between Beijing and the site.

Response: We modified section 3.1.3 in the results and discussions added new lines in the revised MS. Following the reviewer's suggestions we added new correlation table (Table 2) in the revised MS. We have added a new figure (Figure 2b) showing the day and night time wind direction in the Mangshan site. Please see the revised MS.

After deeply digging the dataset we found some intresting facts about the mannitol contribution in the Mangshan aerosol samples. We added several lines in the revised manuscript. Please see lines 358 – 372 in the revised MS.

A major limitation is the lack of any air mass trajectory analysis. The authors state that there was a typical diurnal pattern to the wind direction, with daytime winds from the south and the large cities, and nighttime winds from more rural areas to the north. It would help the reader to see how consistent this pattern was in order to evaluate some of the claims made. I suggest at least a time series plot of wind speed and wind direction, perhaps as a sub-plot to Figure 2, or a wind rose diagram. A couple of representative air mass back trajectories could be instructive as well.

Response: Following the reviewers suggestions we added new figure (Figure 2b). Figure 2 shows the wind direction and magnitude of wind directions in the Mangshan site. Please see the revised MS.

The results section should have more description of the present data set, with more of the previous observations moved to the Introduction. In particular, the description of the PMF results could be expanded, perhaps with a figure showing the time series of the PMF factors.

Response: We significantly modified the result and discussion sections by adding new phrases and relocating the discussions on previous studies. Please see lines 211–214, 242–250, 261–264, 294–297, 306–317, 328–334, 345–347, 353–363, 368–375, 378–384.

We added information about the PMF analysis in the revised MS. Please see lines 187 – 201, 389 – 407, 418 – 429, 442 – 445, 453 – 457, 460 – 462 in the revised MS.

We added Figure S-1 and S-2 as supporting information in the revised MS.

Figure S-1. The Scatter plots between observed (input data) and predicted (modeled data) concentrations show statistical parameter (coefficient of determination (r), Intercept, and Slope) with linear equitation of individual sugar compounds. A blue 1:1 line is provided on this plot for reference (a perfect fit would line up exactly on this line), and the regression line is shown as a dotted red line.

Figure S-2. The time series plots between observed (input data) and predicted (modeled data) concentrations of individual sugar compounds. Blue and red lines show observed (input data) and predicted (modeled data) concentrations, respectively.

Throughout the manuscript, the authors should to be more careful in their descriptions of how the SCs get into the atmosphere. They frequently state that a process or an organism "emits" a sugar compound, which reads somewhat ambiguously (i.e. are gas phase compounds being released?). I think it's OK to use this language, but there first needs to be a clear statement in the Introduction about how SCs get into the atmosphere, or at least the state of the science on that question. How are SCs are released into the air, as fragments of organisms, as whole fungal spores, as individual molecules, etc?

Response: We added new paragraph in the introduction section, including a clear statement on the sources of individual sugar species in the atmosphere. Please see lines 55 – 61, 67 – 75, 78 – 100.

We modified the sentences where we used "emits" in the revised MS.

A further issue that needs to be resolved before this is publishable is the extremely sparse description of the methods. Specific questions are raised below.

At this point this manuscript is essentially a descriptive account of measurements made at a particular location, with some interpretive claims that seem a bit ambiguous. The measurements themselves are of value, but I think for publication, it needs 1) a much more thorough method description section and 2) either a) a scale-back of the claims made, or b) additional analysis in support of the claims.

Response: We significantly modified the sections of materials and methods by including new information and new section in the revised MS. Please see lines 138 - 144, 147 – 149, 161 – 201.

Specific comments:

There are several grammatical issues of subject-verb agreement and lack of pluralization throughout the manuscript. They don't usually impede understanding, but the manuscript would benefit from a thorough grammar check.

Response: We significantly corrected grammatical issues in the manuscript. Please see the revised MS.

It would be very helpful to view the data in Figure 2 directly as a part of Figure 3.

Response: We added new figure in the (Figure 2b) in the revised MS. We also separated figure 3 in three parts (Fig 4a-c, 5a-d and 6a-c) according to the groups of sugar compounds. The new arrangement is very easy to understand and clearer than the earlier version. Please see the revised MS.

Line 66: "SCs are emitted from algae, microbes, pollen, suspended soil particle[s], and associated biota into the atmosphere" This statement reads to me a bit like sugar compounds might be released into the air as individual gas phase molecules, which I don't think is the case (?). Maybe it could be phrased "SCs are emitted as part of aerosols formed from algae..."? Same thing at line 72: I don't think mannitol and arabitol are mostly emitted as individual gases, but are a part of fungal material that gets into the atmosphere. What is physically meant here needs to be a little clearer. Are these SCs usually part of biological fragments, or is this unknown?

Response: Sentence rephrased please see lines 78 – 81 and 85 - 87 in the revised MS.

Section 2.2:

Please provide more details of the sampling apparatus and methodology. Were these samples collected with a high-volume sampler? Where was it installed specifically, and at what height? What aerosol sizes were collected? Were there any measures to avoid the sampling of gas phase components? Did the 3-hr and 9-hr samples overlap in time? What times were the samples started? Perhaps a table of sample collection times would be helpful.

Response: We added Table S1 that contains the detailed description of the sampling procedures and collections. Other information's are added in the text. Please see section "2.1. Site description and aerosol sample collection" in the revised MS.

The twelve (n = 12) aerosol samples on 15 to 18, September and 28 September to 5 October were collected for 9-h from 9 to 18 h. While twenty six (n = 26) aerosol samples of 19 to 27 september were collected from 9 to 12, 12 to 15, 15 to 18 h.

We did not use a denuder system to remove gaseous components in high volume air sampler.

Furthermore, what were the methods for determining the WSOC and total OC? How was Ca(2+) concentration determined? Was the filter cut into sections for each analysis?

Response: Information's are added in the revised manuscript. Please see lines 185 – 186 in the revised MS.

Line 132: What is meant by C13 n-alkane? Is this an isotope standard of one n-alkane? Which one? Or do you mean C13H28, n-tridecane?

Response: $C_{13}$ n-alkane is n-tridecane ($C_{13}H_{28}$).

Line 157: "Hence, it is evident that increased BB activities at nighttime are associated with cool temperature (Fig. 2)." Is this saying that because it's cool at night, it makes sense that there's more BB aerosol at night? Isn't it equally likely that the different air mass origins in the day and at night are the reason?

Response: The sentence is rephrased. Please see lines 242 - 243 in the revised MS. The day-night time difference of the air masses can also influence the concentrations of BB tracers. We added related text please see lines 242 - 250 in the revised MS.

Line 209: "the meteorological conditions". Is this referring to the strong daytime winds and convective activity? It would be clearer to state that directly.

Response: Modified. Please see lines 272 - 274 in the revised MS.

Line 254: "northeasterly (99.5%)". Does this mean 99.5% of the nighttime hours the wind was northeasterly? Please clarify in the text.

Response: Yes, sentence rephrased (Please see lines 308-310 in revised MS).

Line 255: What would cause sugar emissions to decrease with lower temperature? Is there supporting literature for this?

Response: The daytime abmient temperature and solar radiations significantly affect plant activities and, subsequently, emissions of sugar enriched plant fragments. Therefore, the contribution of primary sugars at night was lower than in daytime. Miyazaki et al. (2012) reported the emissions of sugar compounds associeated with light and ambient temoerature at forest site. Please see lines 311 – 317 in the revised MS.

Line 268: Trehalose paragraph. Trehalose didn't show a strong diurnal cycle, but the authors point out a correlation between trehalose and mannitol and arabitol at night, and between trehalose and Ca (2+) in the day. It would be helpful to at least report the corresponding correlation coefficients for the day and night, respectively, for comparison, and possibly to include the corresponding figures in Figure 5.

Response: We added correlation coefficient values in the text. We also added a new Table (Table 2) for correlation of sugar species and meteorological parameters in the manuscript. Please see lines 332 – 334, 336 - 337 in the revised MS.

Line 272: Why would nighttime low RH and temp cause microorganisms to emit more trehalose? Please cite a reference. Again, the use of "emit" here can be confusing. Is it the release of spores that prefers these conditions?

Response: Several studies have reported that the meteorological conditions, i.e., high RH and low temperature, are favorable for the microbes and fungi to discharge spores. The high RH and low temperature was recorded at nigh time therefore the microbes emits spores at night. We added information in the revised manuscript. Please see lines 328-339 in the revised MS. References are also added in the text and references section.

Line 310: Aren't the Mt. Tai measurements higher than Mangshan, not lower?

Response: We deleted this comparison. Please see in the revised MS.

Lines 315-319: I don't understand the reasoning here. How does RH relate to transport from megacities as an explanation for fungal aerosol?

Response: Rephrased. Please see lines 368 – 375, 378-385 in the revised MS.

Line 350: Separate the Factor 3 and Factor 4 descriptions into separate paragraphs.

Response: Factor 3 and Factor 4 are described into separate paragraphs (Please see line 440 – 457 in revised MS).

Line 352: "The PMF results are very well supported by the fact that anhydrosugars are associated with BB in the Mangshan site." Is this referring to results from a previous study? Please cite it.

Response: We modified the phases. Please see lines 448-449 in the revised MS.

Line 410: "Our results also denote that secondary production of OC and WSOC from BB-derived organic precursors was crucial during nighttime at the Mangshan site." What evidence shows this? And do you mean that organic compounds went through chemical changes to form aerosol OC and WSOC, or simply that organics produced during biomass burning were incorporated into aerosol after the burning? In either case, I don't see how we know that.

Response: We deleted the sentence.

**Anonymous Referee #2**

Verma et al. discuss observations of sugars found in the aerosol-phase collected for 1 month in a forested site north of Beijing. The aerosol were collected onto filters and analyzed for the sugars. The authors then describe the pattern of the various sugars throughout the study period and speculate the sources via differences in day- and night-time mass concentration, wind patterns, and PMF. They discuss 5 potential sources, including biomass burning, vegetation, microbial and soil dust, pollen, and fungal.

The results presented here may be of interest to the audience and its scope generally fits within a measurement report. However, along with the concerns discussed by Reviewer #1, the authors nee to address the comments and concerns presented below prior to consideration for publication in ACP.

Major: (1) Statistics: Throughout the text, the authors state that the results are statistically different. However, conducting the t-test with the mean and standard deviation values listed in the table, majority of the observations are statistically similar at the 95% confidence interval and not statistically different. The lack of statistical difference in the observations makes many of the statements the authors use to differentiate day/night and thus sources less substantiated. Further, the correlations shown by the authors in Fig. 5 have very low R values (as stated throughout the text) and suggest that many of the correlations only explain 50% or less of the mass concentration.

Response: We found a positive correlation between sucrose and ambient temperature ($r = 0.52$), sucrose and solar radiation ($r = 0.55$), mannitol and RH ($r = 0.57$), trehalose and arabitol ($r=0.58$), trehalose and mannitol ($r=0.58$), with significance levels <0.05. Therefore we mentioned those positive linear correlation values in the text and discussed accordingly.

(2) Contextualization of results: I agree with Reviewer #1 that the listing of numbers from prior results makes it difficult to understand the conclusions in each section and the whole paper. Further, as highlighted with point (1) above, the data not being statistically different makes sections 3.1.1 thru 3.1.3 very long and repetitive. Also, the listing of numbers from prior studies to ascribe sources for the sugars makes the source apportionment very uncertain. This is also relevant for Section 3.5, where they found no differences in the levoglusoan/mannosan ratio and spend 1.5 pages on this. If this is important, it could be summarized in one paragraph at most.

Response: We deleted repeated sentences, several comparisons with previous studies and significantly modified section 3.1.1. to 3.1.3. We also shorten section 3.5. Please see section 3.1.1 – 3.1.3 and 3.5 in the revised MS.

(3) Methods: Reviewer #1 highlighted many of the methods that should be discussed in more detail. Further, PMF needs to be described in more detail to understand how the 5 results were determined (e.g., how many solutions were there allowed to be, how did the time series look, were the results compared against and investigated against external variables, etc.). Also, agree with Reviewer #1 in how were WSOC, OC, Ca2+, etc determined.

Response: Authors thank the reviewer's valuable comments and suggestions. We make significant changes in the manuscript especially in the section 2 (Materials and methods), and section 3 (Results and discussions). Please see the revised MS.

We added information about the PMF analysis in the revised MS. Please see lines 187 – 201, 389 – 407, 418 – 429, 442 – 445, 453 – 457, 460 – 462 in the revised MS.

We added Figure S-1 and S-2 as supporting information in the revised MS.

Figure S-1. The scatter plots between observed (input data) and predicted (modeled data) concentrations show statistical parameter (coefficient of determination (r), Intercept, and Slope) with linear equitation of individual sugar compounds. (A blue 1:1 line is provided on this plot for reference (a perfect fit would line up exactly on this line), and the regression line is shown as a dotted red line).

Figure S-2. The time series plots between observed (input data) and predicted (modeled data) concentrations of individual sugar compounds. Blue and red lines are shown for observed (input data) and predicted (modeled data) concentrations, respectively.

(4) PMF: I think this is the more interesting and compelling part of the paper. I highly recommend the authors spend more time expanding on this section while reducing the discussion in the other sections. As highlighted above, there are statistical concerns, thus shortening them while increasing the discussion about PMF, which had lower statistical concern.

Response: We added more sentences in the PMF analysis section. Please see line 386 – 468 in the revised MS.

(5) Figures: The x-axis/date is very hard to read in all figures. It is unclear what the values are shown in different colors in Fig. 3.

Response: We separated Figure 3 as Figures 4a-c, 5a-d and 6a-c. We also modified x-axis in the manuscript and changes are added in the figure captions.

Minor: Please review the grammar throughout the paper, as highlighted by Reviewer #1.

Response: According to the reviewer's suggestion, whole manuscript is properly checked for the grammatical mistakes.

Fig. 2.

[Figure]

Fig. 4.

[Figure]

Fig. 5.

[Figure]

Fig. 6.

[Figure]

**Table 2.** Statistical summary of correlations among the chemical species and meteorological variables in aerosol samples collected at a forest site in northern Japan.

| Linear regression | Correlation coefficient | p value | Significance of correlation at P value < 0.05 |
|---|---|---|---|
| **Overall (n = 58)** | | | |
| Levoglucosan vs. Galactosan | 0.98 | < 0.05 | Significant |
| Levoglucosan vs. Mannosan | 0.97 | < 0.05 | Significant |
| Mannosan vs. Galactosan | 0.98 | < 0.05 | Significant |
| Sucrose vs. Temperature | 0.52 | < 0.05 | Significant |
| Sucrose vs. Solar radiation | 0.55 | < 0.05 | Significant |
| Arabitol vs. Mannitol | 0.81 | < 0.05 | Significant |
| Arabitol vs. RH | 0.69 | < 0.05 | Significant |
| Mannitol vs. RH | 0.57 | < 0.05 | Significant |
| Glucose vs. Fructose | 0.94 | < 0.05 | Significant |
| Trehalose vs. Arabitol | 0.58 | < 0.05 | Significant |
| Trehalose vs. Mannitol | 0.58 | < 0.05 | Significant |
| Trehalose vs. $Ca^{2+}$ | 0.70 | < 0.05 | Significant |
| **Daytime (n = 38)** | | | |
| Sucrose vs. $Ca^{2+}$ | 0.32 | > 0.05 | Not significant |
| Glucose vs. $Ca^{2+}$ | 0.02 | > 0.05 | Not significant |
| Trehalose vs. Arabitol | 0.49 | < 0.05 | Significant |
| Trehalose vs. Mannitol | 0.51 | < 0.05 | Significant |
| Trehalose vs. $Ca^{2+}$ | 0.81 | < 0.05 | Significant |
| Fructose vs. Mannitol | 0.79 | < 0.05 | Significant |
| Levoglucosan vs. OC | 0.45 | < 0.05 | Significant |
| Levoglucosan vs. WSOC | 0.40 | < 0.05 | Significant |
| **Nighttime  (n = 20)** | | | |
| Sucrose vs. $Ca^{2+}$ | 0.37 | > 0.05 | Not significant |
| Glucose vs. $Ca^{2+}$ | 0.27 | > 0.05 | Not significant |
| Trehalose vs. Arabitol | 0.76 | < 0.05 | Significant |
| Trehalose vs. Mannitol | 0.85 | < 0.05 | Significant |
| Trehalose vs. $Ca^{2+}$ | 0.61 | < 0.05 | Significant |
| Fructose vs. Mannitol | 0.86 | < 0.05 | Significant |
| Levoglucosan vs. OC | 0.81 | < 0.05 | Significant |
| Levoglucosan vs. WSOC | 0.70 | < 0.05 | Significant |

**Supporting Information**

Table S-1 Sampling informations.

Sampler:High Volume Air Sampler (Kimoto-AS810A)

| Sample ID | Start Time | Finish Time | Center time | Total Time (Hours) | Total Air V (m$^3$) | Filter Total (cm$^2$) |
|---|---|---|---|---|---|---|
| CHN-131 | 2007/9/15 9:08 | 2007/9/15 17:57 | 15/09/07 13:32 | 8:49 | 533.25 | 405.3 |
| CHN-132 | 15/09/07 18:01 | 16/09/07 8:25 | 16/09/07 1:13 | 14:24 | 873.09 | 405.3 |
| CHN-133 | 16/09/07 8:29 | 16/09/07 17:53 | 16/09/07 13:11 | 9:24 | 563.03 | 405.3 |
| CHN-134 | 16/09/07 17:57 | 17/09/07 8:56 | 17/09/07 1:26 | 14:59 | 907.48 | 405.3 |
| CHN-135 | 17/09/07 9:00 | 2007/9/17 17:38 | 17/09/07 13:19 | 8:38 | 515.44 | 405.3 |
| CHN-136 | 18/09/07 9:58 | 18/09/07 18:03 | 18/09/07 14:00 | 8:05 | 490.5 | 405.3 |
| CHN-137 | 2007/9/18 18:06 | 19/09/07 8:54 | 19/09/07 1:30 | 14:48 | 905.69 | 405.3 |
| CHN-139 | 19/09/07 9:04 | 19/09/07 12:00 | 19/09/07 10:32 | 2:56 | 177.5 | 405.3 |
| CHN-140 | 19/09/07 12:04 | 19/09/07 15:01 | 19/09/07 13:32 | 2:57 | 180.14 | 405.3 |
| CHN-141 | 19/09/07 15:05 | 19/09/07 18:01 | 19/09/07 16:33 | 2:56 | 179.44 | 405.3 |
| CHN-142 | 19/09/07 18:04 | 20/09/07 9:01 | 20/09/07 1:32 | 14:57 | 920.18 | 405.3 |
| CHN-143 | 20/09/07 9:05 | 20/09/07 12:02 | 20/09/07 10:33 | 2:57 | 172.93 | 405.3 |
| CHN-144 | 2007/9/20 12:06 | 2007/9/20 15:01 | 20/09/07 13:33 | 2:55 | 177.25 | 405.3 |
| CHN-145 | 20/09/07 15:04 | 20/09/07 17:59 | 20/09/07 16:31 | 2:55 | 178.6 | 405.3 |
| CHN-146 | 20/09/07 18:02 | 21/09/07 8:57 | 21/09/07 1:29 | 14:55 | 919.91 | 405.3 |
| CHN-147 | 21/09/07 9:00 | 21/09/07 12:02 | 21/09/07 10:31 | 3:02 | 180.96 | 405.3 |
| CHN-149 | 21/09/07 15:06 | 21/09/07 17:59 | 21/09/07 16:32 | 2:53 | 177.52 | 405.3 |
| CHN-150 | 21/09/07 18:02 | 22/09/07 9:00 | 22/09/07 1:31 | 14:58 | 915.67 | 405.3 |
| CHN-151 | 22/09/07 9:02 | 22/09/07 11:59 | 22/09/07 10:30 | 2:57 | 178.6 | 405.3 |
| CHN-152 | 22/09/07 12:02 | 22/09/07 14:59 | 22/09/07 13:30 | 2:57 | 177.73 | 405.3 |
| CHN-153 | 22/09/07 15:02 | 22/09/07 17:58 | 22/09/07 16:30 | 2:56 | 180.68 | 405.3 |
| CHN-154 | 22/09/07 18:01 | 23/09/07 8:56 | 23/09/07 1:28 | 14:55 | 911.27 | 405.3 |
| CHN-155 | 23/09/07 9:00 | 23/09/07 11:57 | 23/09/07 10:28 | 2:57 | 177.82 | 405.3 |
| CHN-156 | 2007/9/23 12:00 | 23/09/07 14:59 | 23/09/07 13:29 | 2:59 | 179.3 | 405.3 |
| CHN-157 | 23/09/07 15:02 | 23/09/07 17:59 | 23/09/07 16:30 | 2:57 | 180.75 | 405.3 |
| CHN-158 | 23/09/07 18:03 | 24/09/07 8:55 | 24/09/07 1:29 | 14:52 | 918.89 | 405.3 |
| CHN-159 | 24/09/07 8:59 | 24/09/07 11:56 | 24/09/07 10:27 | 2:57 | 178.16 | 405.3 |
| CHN-160 | 24/09/07 11:59 | 24/09/07 14:57 | 24/09/07 13:28 | 2:58 | 180.62 | 405.3 |
| CHN-161 | 2007/9/24 15:00 | 24/09/07 17:56 | 24/09/07 16:28 | 2:56 | 179.84 | 405.3 |
| CHN-163 | 24/09/07 18:11 | 25/09/07 9:01 | 25/09/07 1:36 | 14:50 | 899.43 | 405.3 |
| CHN-164 | 25/09/07 9:04 | 25/09/07 12:07 | 25/09/07 10:35 | 3:03 | 184.92 | 405.3 |
| CHN-165 | 25/09/07 12:09 | 25/09/07 14:58 | 25/09/07 13:33 | 2:49 | 169.63 | 405.3 |
| CHN-166 | 25/09/07 15:01 | 25/09/07 17:58 | 25/09/07 16:29 | 2:57 | 180.45 | 405.3 |
| CHN-167 | 25/09/07 18:01 | 26/09/07 9:01 | 26/09/07 1:31 | 15:00 | 912.34 | 405.3 |
| CHN-168 | 2007/9/26 9:04 | 26/09/07 11:59 | 26/09/07 10:31 | 2:55 | 176.33 | 405.3 |
| CHN-169 | 26/09/07 12:02 | 26/09/07 14:58 | 26/09/07 13:30 | 2:56 | 176.46 | 405.3 |
| CHN-170 | 26/09/07 15:01 | 26/09/07 17:58 | 26/09/07 16:29 | 2:57 | 181.09 | 405.3 |
| CHN-171 | 26/09/07 18:01 | 27/09/07 8:56 | 27/09/07 1:28 | 14:55 | 916.56 | 405.3 |
| CHN-172 | 27/09/07 8:59 | 27/09/07 12:00 | 27/09/07 10:29 | 3:01 | 183.05 | 405.3 |
| CHN-173 | 27/09/07 12:03 | 27/09/07 15:01 | 27/09/07 13:32 | 2:58 | 180.88 | 405.3 |
| CHN-174 | 27/09/07 15:04 | 27/09/07 17:59 | 27/09/07 16:31 | 2:55 | 179.04 | 405.3 |
| CHN-175 | 27/09/07 18:02 | 28/09/07 9:15 | 28/09/07 1:38 | 15:13 | 936.45 | 405.3 |
| CHN-176 | 28/09/07 9:19 | 28/09/07 17:57 | 28/09/07 13:38 | 8:38 | 519.1 | 405.3 |
| CHN-178 | 28/09/07 18:10 | 29/09/07 8:59 | 29/09/07 1:34 | 14:49 | 907.18 | 405.3 |
| CHN-179 | 29/09/07 9:01 | 29/09/07 17:59 | 29/09/07 13:30 | 8:58 | 539.66 | 405.3 |
| CHN-180 | 29/09/07 18:01 | 30/09/07 8:57 | 30/09/07 1:29 | 14:56 | 922.34 | 405.3 |
| CHN-181 | 30/09/07 9:00 | 30/09/07 17:57 | 30/09/07 13:28 | 8:57 | 531.89 | 405.3 |
| CHN-182 | 30/09/07 18:00 | 01/10/07 8:55 | 01/10/07 1:27 | 14:55 | 896.33 | 405.3 |
| CHN-183 | 01/10/07 8:58 | 01/10/07 17:54 | 01/10/07 13:26 | 8:56 | 538.04 | 405.3 |
| CHN-184 | 01/10/07 17:57 | 02/10/07 8:56 | 02/10/07 1:26 | 14:59 | 926.33 | 405.3 |
| CHN-185 | 02/10/07 8:59 | 02/10/07 17:56 | 02/10/07 13:27 | 8:57 | 531.43 | 405.3 |
| CHN-186 | 02/10/07 17:59 | 03/10/07 8:59 | 03/10/07 1:29 | 15:00 | 913.14 | 405.3 |
| CHN-187 | 03/10/07 9:01 | 03/10/07 17:52 | 03/10/07 13:26 | 8:51 | 523.19 | 405.3 |
| CHN-188 | 03/10/07 17:54 | 04/10/07 8:57 | 04/10/07 1:25 | 15:03 | 919.84 | 405.3 |
| CHN-189 | 04/10/07 9:00 | 04/10/07 17:56 | 04/10/07 13:28 | 8:56 | 535.79 | 405.3 |
| CHN-190 | 04/10/07 18:00 | 05/10/07 8:59 | 05/10/07 1:29 | 14:59 | 918.46 | 405.3 |
| CHN-191 | 05/10/07 9:04 | 05/10/07 17:56 | 05/10/07 13:30 | 8:52 | 535.52 | 405.3 |
| CHN-192 | 05/10/07 17:59 | 06/10/07 8:53 | 06/10/07 1:26 | 14:54 | 906.91 | 405.3 |
| CHN-138 | 19/09/07 9:00 | blank | 19/09/07 9:00 | | | 405.3 |
| CHN-148 | 21/09/07 13:52 | blank | 21/09/07 13:52 | | | 405.3 |
| CHN-162 | 24/09/07 17:58 | blank | 24/09/07 17:58 | | | 405.3 |
| CHN-177 | 28/09/07 18:00 | blank | 28/09/07 18:00 | | | 405.3 |

[Figure]

Figure S-1. The Scatter plots between observed (input data) and predicted (modeled data) concentrations show statistical parameter (coefficient of determination (r), Intercept, and Slope) with linear equitation of individual sugar compounds. (A blue 1:1 line is provided on this plot for reference (a perfect fit would line up exactly on this line), and the regression line is shown as a dotted red line).

[Figure]

Figure S-2. The time series plots between observed (input data) and predicted (modeled data) concentrations of individual sugar compounds. (A blue line and redline shown as observed (input data) and predicted (modeled data) concentrations, respectively.

[revised manuscript text omitted]

List of all relevant changes
1. We add three key points in the revised MS.
2. We added new paragraph in the introduction section.
3. The materials and methods sections were modified by adding new information and subsection.
4. We modified the discussion section by re-organizing paragraphs.
5. We added new figure as figure 2b and moved figure 4 to Figure 3.
6. We split figure 3 into three Figures as Figure 4 (anhydrosugars), Figure 5 (primary sugars), and Figure 6 (sugar alcohols). We added the modified version of Figure 9. We changed Figure 5 to table 2.
7. We added table S-1 (sampling information), Figure S-1 (PMF results), and Figure S-2 (PMF results) as supplementary material.
8. We went through the paper and corrected the typos. We also rewrote sentences that were difficult to read.

[revised manuscript text omitted]

Fig. 4

[Figure]

Fig. 5.

[Figure]

Fig. 6.

[Figure]

Fig. 7.

[Figure]

Fig. 8.

[Figure]

Fig. 9.

[Figure]

[Figure]

Fig. 10.

[Figure]

Table 1. Minimum, maximum, average and standard deviations of concentrations of sugar compounds in aerosol samples (TSP) from Mangshan, China.

| Sugar Compounds | Overall | | | | Daytime (n = 38) | | | | Nighttime (n = 20) | | | |
|---|---|---|---|---|---|---|---|---|---|---|---|---|
| | Min | Max | Avg. | S.D. | Min | Max | Avg. | S.D. | Min | Max | Avg. | S.D. |
| **Anhydrosugars** | | | | | | | | | | | | |
| Galactosan | 0.14 | 48.0 | 10.1 | 11.9 | 0.14 | 45.3 | 8.53 | 10.5 | 0.69 | 48.0 | 13.0 | 14.0 |
| Mannosan | 0.13 | 26.1 | 6.05 | 6.33 | 0.13 | 24.3 | 5.37 | 6.01 | 0.53 | 26.1 | 7.35 | 6.87 |
| Levoglucosan | 1.17 | 482 | 100 | 119 | 1.17 | 418 | 83.2 | 106 | 5.66 | 482 | 132 | 138 |
| **Sugar alcohols** | | | | | | | | | | | | |
| Arabitol | 3.89 | 72.2 | 29.1 | 21.5 | 3.99 | 72.2 | 32.5 | 22.0 | 3.89 | 71.3 | 22.5 | 19.4 |
| Mannitol | 4.19 | 182 | 44.1 | 34.5 | 4.19 | 182 | 51.7 | 37.5 | 4.40 | 87.7 | 29.6 | 22.3 |
| Inositol | 0.23 | 6.8 | 2.62 | 1.81 | 0.27 | 6.80 | 3.14 | 1.90 | 0.23 | 4.65 | 1.62 | 1.09 |
| **Primary sugars** | | | | | | | | | | | | |
| Fructose | 1.72 | 177 | 20.1 | 24.6 | 1.72 | 177 | 23.9 | 29.3 | 2.64 | 30.9 | 12.8 | 7.67 |
| Glucose | 1.86 | 297 | 40.0 | 43.4 | 1.86 | 297 | 44.2 | 50.8 | 4.52 | 108 | 32.0 | 22.8 |
| Sucrose | 0.02 | 474 | 58.5 | 96.5 | 0.02 | 474 | 82.9 | 112 | 0.04 | 60.1 | 12.3 | 15.1 |
| Trehalose | 0.06 | 39.5 | 14.3 | 10.5 | 0.06 | 34.9 | 15.3 | 10.6 | 0.87 | 39.5 | 12.3 | 10.2 |
| Anhydrosugars | 6.01 | 556 | 116 | 137 | 6.01 | 476 | 97.1 | 122 | 6.88 | 556 | 152 | 159 |
| Primary sugars | 9.41 | 565 | 133 | 125 | 9.41 | 565 | 166 | 141 | 10.5 | 172 | 69.4 | 43.0 |
| Sugar alcohols | 8.53 | 259 | 75.8 | 54.7 | 9.09 | 259 | 87.4 | 57.5 | 8.53 | 164 | 53.7 | 41.9 |
| Total Sugars | 30.8 | 875 | 325 | 232 | 34.1 | 875 | 351 | 240 | 30.8 | 759 | 276 | 212 |
| Anhydrosugars (%) | | | 31.9 | | | | 24.6 | | | | 45.7 | |
| Primary sugars (%) | | | 41.8 | | | | 47.3 | | | | 31.3 | |
| Sugar alcohols (%) | | | 26.4 | | | | 28.1 | | | | 23.0 | |

Table 2. Statistical summary of correlations among the chemical species and meteorological
variables in aerosol samples collected at a forest site in northern Japan

| Linear regression | Correlation coefficient | p value | Significance of correlation at P value < 0.05 |
|---|---|---|---|
| Overall (n = 58) | | | |
| Levoglucosan vs. Galactosan | 0.98 | < 0.05 | Significant |
| Levoglucosan vs. Mannosan | 0.97 | < 0.05 | Significant |
| Mannosan vs. Galactosan | 0.98 | < 0.05 | Significant |
| Sucrose vs. Temperature | 0.52 | < 0.05 | Significant |
| Sucrose vs. Solar radiation | 0.55 | < 0.05 | Significant |
| Arabitol vs. Mannitol | 0.81 | < 0.05 | Significant |
| Arabitol vs. RH | 0.69 | < 0.05 | Significant |
| Mannitol vs. RH | 0.57 | < 0.05 | Significant |
| Glucose vs. Fructose | 0.94 | < 0.05 | Significant |
| Trehalose vs. Arabitol | 0.58 | < 0.05 | Significant |
| Trehalose vs. Mannitol | 0.58 | < 0.05 | Significant |
| Trehalose vs. $Ca^{2+}$ | 0.70 | < 0.05 | Significant |
| Daytime (n = 38) | | | |
| Sucrose vs. $Ca^{2+}$ | 0.32 | > 0.05 | Not significant |
| Glucose vs. $Ca^{2+}$ | 0.02 | > 0.05 | Not significant |
| Trehalose vs. Arabitol | 0.49 | < 0.05 | Significant |
| Trehalose vs. Mannitol | 0.51 | < 0.05 | Significant |
| Trehalose vs. $Ca^{2+}$ | 0.81 | < 0.05 | Significant |
| Fructose vs. Mannitol | 0.79 | < 0.05 | Significant |
| Levoglucosan vs. OC | 0.45 | < 0.05 | Significant |
| Levoglucosan vs. WSOC | 0.40 | < 0.05 | Significant |
| Nighttime (n = 20) | | | |
| Sucrose vs. $Ca^{2+}$ | 0.37 | > 0.05 | Not significant |
| Glucose vs. $Ca^{2+}$ | 0.27 | > 0.05 | Not significant |
| Trehalose vs. Arabitol | 0.76 | < 0.05 | Significant |
| Trehalose vs. Mannitol | 0.85 | < 0.05 | Significant |
| Trehalose vs. $Ca^{2+}$ | 0.61 | < 0.05 | Significant |
| Fructose vs. Mannitol | 0.86 | < 0.05 | Significant |
| Levoglucosan vs. OC | 0.81 | < 0.05 | Significant |
| Levoglucosan vs. WSOC | 0.70 | < 0.05 | Significant |

The data of $Ca^{2+}$, OC and WSOC are adapted from He et al. (2015).

---

## Author Response (AR2)

The authors have made significant additions to the manuscript in the areas recommended, and I think it is now publishable. I think though that some of the new text should get a second look for clarity/grammar before finalizing. I highlight a few sentences here that could be improved:

Response: Authors are thankful to the reviewers for their valuable comments and suggestions.

Line 60 suggestion: "On a global scale, one-fourth of anthropogenic aerosol (mass?) is contributed by China, approximately 70% of which was emitted from coal burning"

Response: Modified. Please see lines 60-61 in the revised MS.

"Globally, significant anthropogenic and carbonaceous aerosols are contributed by China (Cooke et al., 1999, Wang et al., 2007)."

Cooke, W.F., Liousse, C., Cachier, H., and Feichter, J.: Construction of a 1° X 1° fossil fuel emission data set for carbonaceous aerosol and implementation and radiative impact in the ECHAM4 model. J. Geophys. Res. Atmos., 104, 22137–22162, https://doi.org/10.1029/1999JD900187, 1999.

Wang, G., Kawamura, K., Zhao, X., Li, Q., Dai, Z., and Niu, H.: Identification, abundance, and seasonal variation of anthropogenic organic aerosols from a mega-city in China. Atmos. Environ., 41, 407–416, https://doi.org/10.1016/j.atmosenv.2006.07.033, 2007.

Line 65 suggestion: "OAs are composed of a complex…"

Response: Modified. Please see lines 65-66 in the revised manuscript.

"Organic aerosols (OAs) are composed of a complex mixture of diverse molecules (Xu et al., 2011)."

Line 363: suggestion "suggested it had sources in addition to fungal spores…"

Response: Modified. Please see lines 391-392 in revised manuscript.

"In contrast, the higher concentration of mannitol than arabitol suggested it had sources in addition to fungal spores in the Mangshan forest site."

Line 585: "contributions of 36%" specify contributions of SCs if that's meant

Response: Corrected. Please see line 619-621 in the revised MS.

"PMF results concluded the contributions of 36% from vegetation (21% vegetation factor and 15% pollen factor) and 37% from microbial and fungal species (21% microbial soil dust and 16% fungal factor) of total measured SCs."

**Anonymous Referee #2**
Verma et al. reports measurements of sugars collected on filters for about 3 months to determine the potential sources of the sugars. Though the authors addressed many of the comments from both reviewers, there are still concerns in regards to the methods and the interpretation of the data. Some of these concerns were brought up in the first set of reviews but were not addressed at all.

Response: Authors are thankful to the reviewer's valuable comments and suggestions, which help to upgrade the quality of the manuscript. We made significant changes in the manuscript acording to the reviewer's suggestions.

**Major**

(1) It is not clear if they sampled any aerosol size < 100 um vs having a cut-off.

Response: In this sampling, we used a high-volume air sampler (Kimoto-AS810A) to collect total suspended particles (TSP) without cut-off device. We added information in the revised manuscript. Please see lines 146-147 in revised MS.

(2) Since there is no denuder, what is the impact of condensation of semi-volatile gases onto the filters in the analysis? At least levoglucosan is known to partition between the gas and particle phase. This should be addressed as either a potential uncertainty/limitation if unknown or at least discussed if it is known the potential interference/impact it may have on the reported mass.

Response: In this work, no denuder was used to remove semi-volatile species. The removal of the gaseous species in the denuder distorts the gas-particle equilibrium and leads to the dissociation of the particulate phase during the sampling. This phenomenon is particularly true for volatile organic species. It leads to significant errors in determining gas-to-particle partitioning of organic aerosols due to underestimating the particle phase (Dhawan and Biswas et al., 2019). In this study, we reported nonvolatile sugar compounds. However, the levoglucosan partition between the gas and particle phases, but their concentration was low. The sampling time was rather short due to the day and night sampling. Therefore, we believe that the uncertainty due to the gas phases in the particulate species concentration might be insignificant. Sentences are added, please see lines 147-152 in the revised MS.

"In the sampling, no denuder was applied to remove semi-volatile gases because the filter samples were used to analyze nonvolatile sugar compounds. However, the levoglucosan partition between the gas and particle phases, but their concentration was low. The sampling time was rather short due to the day and night sampling.

Therefore, the uncertainty due to the gas phases in the particulate species concentration might be insignificant."

(3) Though the author cite a reference on how OC, WSOC, and $Ca^{2+}$ was measured/determined, the authors should still add a brief description so that readers do not have to dig through other papers. This can either be added to the main paper or SI, but a brief description in how these were measured would be beneficial for the overall paper.

Response: According to the reviewer's suggestions, we added a brief description of methods for the determination of OC, WSOC, and inorganic ions as new section 2.4 in the materials and methods. Please see lines 196-208 in the revised MS.

**2.4. Chemical analyses of organic carbon, water-soluble organic carbon and inorganic ions**

The data set and methods for the determination of organic carbon (OC), water-soluble organic carbon (WSOC) and inorganic ion ($Ca^{2+}$) were reported in He et al. (2015). Briefly, the concentrations of OC were measured using a semi-continuous OC/EC analyzer (Sunset Laboratory Inc., Portland, OR, USA). A punch of the filter ($\Phi$14 mm) was placed in a quartz boat inside the thermal desorption chamber of the analyzer, and then stepwise heating (IMPROVE) was applied. The oven temperature was programmed as follows: under He, every 2 minutes, the oven temperature was increased starting from 250°C for 2 min, at 450°C for 2 min, and at 550°C for 2 min. After that, 550°C was maintained for two minutes under He mixed with 10% $O_2$, then at 700°C for 2 min and at 870°C for 3.5 min. NDIR detector was used to determine $CO_2$ generated in the above process (Wang et al., 2005). The carbon content of the sample that evolves to $CO_2$ between 250 and 700°C was defined as OC.

Aliquots of the filter samples (3.14 $cm^2$) were extracted with Milli Q water for the water-soluble inorganic ion and WSOC measurements. After extraction, one part was used for the analyses of inorganic ions ($SO_4^{2-}$, $NO_3^-$, $Cl^-$, $NH_4^+$, $Na^+$, $Ca^{2+}$, $K^+$ and $Mg^{2+}$) using an ion chromatography (IC) system (761 Compact IC, Metrohm, Switzerland). Cations on a Shodex YK-421 column with 4mM $H_3PO_4$ as eluent and anions were separated on a Shodex SI-90 4E column with 1.8mM $Na_2CO_3$ and 1.7mM $NaHCO_3$ as eluent. The injection loop volume was 200 $\mu$l. Both cations and anions were quantified against a standard calibration curve. Another part of the filtered water extract was acidified with 1.2 M HCl and purged with pure air to

remove dissolved inorganic carbon and volatile organics. Then WSOC was measured with a carbon analyzer (Shimadzu, TOC-5000). Procedural blanks were carried out in parallel with real samples to account for any contamination (He et al., 2015).

(4) It is still not clear how PMF was determined. What software was used? Were the solutions constrained? How were the solutions selected? At minimum, a time series of the solutions should be added.

Response: For the analysis of source apportionment, Positive matrix factorization (PMF) software version 5.0 (Environmental Protection Agency, USA) was used. The information is added in the revised MS, please see lines 424-425.

The additional information about the PMF are added as text in the revised manuscript and the time series plots of the solutions also added as Figure S-3 in suplementory. Please see lines 445 – 450 in the revised MS.

"The time series plots of the factors solutions determined by PMF were similar to the temporal plots of the concentration of sugar species of the factor composition (Figure S-3). The numbers of factors were reduced if the pair of factors was strongly correlated. The composition of each factor was also checked; none of the pair of factors were found with similar composition. We also investigated the change in factor profile with positive and negative values of $f_{peak}$ for the chosen solution in the PMF analysis."

[Figure]

Figure S-3. The time series plots of the factors solutions determined by PMF

(5) As previously mentioned, looking at the average values, the standard deviation, and the number of measurements, many of the "day" "night" differences are not significantly different (student t-test, 95% confidence interval). This impacts the narrative throughout Sect. 3.1.

Response: We discussed the diurnal variations on the basis of difference in the concentrations of sugar species during day and night time, however, we did not observe statistically significant differences (student t-test, 95% confidence interval, $p > 0.05$) in their atmospheric abundances. Therefore, we added few lines in the revied MS. Please see lines 239 and 242.

"The overall concentrations of SCs varied from 30.8–875 ng m$^{-3}$ (avg. 325 ng m$^{-3}$), which was higher in the daytime (315 ng m$^{-3}$) and lower at nighttime (276 ng m$^{-3}$), however, we did not observe statistically significant differences (student t-test, 95% confidence interval, $p > 0.05$) in their atmospheric abundances."

(6) It is still unclear how biomass burning/levoglucosan is "higher" during night vs day (though as mentioned above, there is no statistical difference using a student ttest) as the wind is coming from the "forested" site instead of Beijing. Where is this source of biomass burning coming from?

Response: We have mentioned in the text that the local biomass burning are prime sources for nighttime levoglucosan contribution in Mangshan aerosol. The nighttime samples were collected from 18:00h to 09:00h, including peak hours of BB for domestic purposes. Therefore, it is reasonable to detect higher abundances of BB tracers in the nighttime than daytime. However, northeast wind (99.5%) was dominated at night, coming from the forested site, but they carry relatively clean air masses; therefore, it has no significant levoglucosan contribution at nighttime samples from the forest site. In contrast, the daytime wind carries levoglucosan from Beijing City to the sampling site, which contribut levoglucosan during daytime aerosol samples. Please see sub-section 3.1.1 in the revised MS.

We agreed with the reviewers that day/night differences in the atmospheric abundance of some sugar species are not statistically significant. Therefore we added a line in section 3.1. Please see lines 241-242 in the revised MS.

"However, we did not observe statistically significant differences (student t-test, 95% confidence interval, $p > 0.05$) in their atmospheric abundances."

**Minor**

(1) It is beneficial to type exactly what you have changed in the responses to reviewers instead of asking them to dig through the document to find the changes. Also, it is unclear what, if any, sentences/sections were removed as only additions are highlighted.

Response: We apologize if reviewers feel some difficulties to read response letter. According to the reviewer's comment, we have included all the changes made in the revised manuscript into the response letter.

(2) Line 62 - 66: Sources of OA are misrepresented here, as majority of the OA originate from the photoxidation of gases into SOA (e.g., Jimenez et al, Science, 2009).

Response: Modified. Please see lines 61-65 in the revised manuscript.

"Beijing is one of the largest polluted cities in East Asia; its air quality deteriorates seriously due to massive emissions of anthropogenic aerosols from vehicles and

industries (Cao et al., 2014; Qiao et al., 2018; Tao et al., 2017; Wei et al., 2018; Yu et al., 2013)."

(3) Line 170: Was the mass spec a quad or TOF? What was the resolution?
Response: The quadrupole mass spectrometer with resolution of 1000 was used in this study.

(4) Fig. 5: I would recommend making the y-axis label that corresponds to $Ca^{2+}$ the same color as it can be hard to interpret which axis corresponds to which data. Also, $Ca^{2+}$ is not included in the caption.
Response: We change the color of y-axis, and information added in the figure caption.

"Y-axis shows temporal variations in the concentrations ($\mu g\ m^{-3}$) of $Ca^{2+}$."